# Small RNA-binding protein RapZ mediates cell envelope precursor sensing and signaling in *Escherichia coli*

Muna A Khan[1], Svetlana Durica-Mitic[1], Yvonne Göpel[1], Ralf Heermann[2] & Boris Görke[1,*] (ID)

## Abstract

The RNA-binding protein RapZ cooperates with small RNAs (sRNAs) GlmY and GlmZ to regulate the *glmS* mRNA in *Escherichia coli*. Enzyme GlmS synthesizes glucosamine-6-phosphate (GlcN6P), initiating cell envelope biosynthesis. GlmZ activates *glmS* expression by base-pairing. When GlcN6P is ample, GlmZ is bound by RapZ and degraded through ribonuclease recruitment. Upon GlcN6P depletion, the decoy sRNA GlmY accumulates through a previously unknown mechanism and sequesters RapZ, suppressing GlmZ decay. This circuit ensures GlcN6P homeostasis and thereby envelope integrity. In this work, we identify RapZ as GlcN6P receptor. GlcN6P-free RapZ stimulates phosphorylation of the two-component system QseE/QseF by interaction, which in turn activates *glmY* expression. Elevated GlmY levels sequester RapZ into stable complexes, which prevents GlmZ decay, promoting *glmS* expression. Binding of GlmY also prevents RapZ from activating QseE/QseF, generating a negative feedback loop limiting the response. When GlcN6P is replenished, GlmY is released from RapZ and rapidly degraded. We reveal a multifunctional sRNA-binding protein that dynamically engages into higher-order complexes for metabolite signaling.

**Keywords** cell envelope precursor glucosamine-6-phosphate; negative feedback loop; RNA-binding protein RapZ; small RNAs GlmY and GlmZ; two-component system QseE-QseF

**Subject Categories** Microbiology, Virology & Host Pathogen Interaction; RNA Biology

**The EMBO Journal (2020) 39: e103848**

## Introduction

Post-transcriptional regulation mediated by RNA and RNA-binding proteins (RBPs) has emerged as critical layer in regulation of gene expression and cellular physiology in all organisms. Bacteria, which are frequently challenged with altered environmental conditions, make extensive use of small regulatory RNAs (sRNAs) to achieve rapid adaptation (Storz *et al*, 2011; Wagner & Romby, 2015). sRNAs do not act alone, but frequently function in conjunction with RBPs. Global approaches revealed a plethora of new RBPs and RBP-RNA interactions in eukaryotes, often involving unconventional RNA-binding domains (RBDs; Hentze *et al*, 2018). In contrast, a few RBPs cooperating with sRNAs are known in bacteria (Holmqvist & Vogel, 2018; Babitzke *et al*, 2019), albeit ~6–7% of a typical bacterial proteome may feature RNA-binding activity (Ghosh *et al*, 2019). Hfq, ProQ, and CsrA emerged as global RBPs governing large post-transcriptional networks, either by facilitating the activities of base-pairing sRNAs (Holmqvist *et al*, 2018; Santiago-Frangos & Woodson, 2018) or by acting as pleiotropic mRNA repressor (Potts *et al*, 2017), but little is known beyond. It also remains largely unclear how sRNA-binding proteins are themselves regulated, how they are embedded in the protein–protein interaction network, and to which extent they cross-talk with transcriptional regulators.

Protein RapZ (32.49 kDa; formerly YhbJ) in *Escherichia coli* represents a novel type of RBP that was originally identified by the phenotype of chronic overproduction of enzyme GlmS in corresponding mutants (Kalamorz *et al*, 2007). GlmS synthesizes glucosamine-6-phosphate (GlcN6P), the starting metabolite for cell envelope synthesis. RapZ was found to promote decay of a dedicated sRNA by a global RNase (Göpel *et al*, 2013), an activity also observed in other sRNA circuits (Leng *et al*, 2016), revealing a mechanism allowing for programmed turnover of a particular transcript. The *trans*-encoded sRNA GlmZ activates *glmS* mRNA translation by base-pairing (Kalamorz *et al*, 2007; Urban & Vogel, 2008). GlmZ is inactivated through processing by endoribonuclease RNase E in the base-pairing region. Cleavage depends on the "adaptor" function of RapZ, which binds GlmZ at its central stem loop and recruits RNase E by interaction with its catalytic domain (Göpel *et al*, 2013, 2016). RapZ forms a swapped dimer of dimers, an assembly required for activity (Gonzalez *et al*, 2017). Possibly, RapZ and the likewise tetrameric RNase E catalytic domain sandwich GlmZ for cleavage in an encounter complex. The RapZ protomer consists of two globular domains, of which the C-terminal domain (CTD) can dimerize and bind RNA on its own. The RBD is apparently formed by the surface-exposed 19 residues long C-terminal tail enriched in positive charges and SR motifs (Göpel *et al*,

1   Department of Microbiology, Immunobiology and Genetics, Max Perutz Labs, Vienna Biocenter (VBC), University of Vienna, Vienna, Austria
2   Microbiology and Wine Research, Institute for Molecular Physiology, Johannes Gutenberg-University Mainz, Mainz, Germany
    *Corresponding author. Tel: +43 1 427754603; E-mail: boris.goerke@univie.ac.at

2013; Gonzalez *et al*, 2017), features also present in emerging non-typical eukaryotic RBDs (Hentze *et al*, 2018). The RapZ-CTD exhibits structural homology to 6-phosphofructokinase (Gonzalez *et al*, 2017), re-emphasizing a recently recognized relationship between metabolic enzymes and RNA-binding activity (Hentze *et al*, 2018).

The adaptor function of RapZ is controlled by GlcN6P through a mechanism that involves the decoy sRNA GlmY. Albeit homologous to GlmZ, GlmY lacks the base-pairing site and cannot directly regulate *glmS* (Reichenbach *et al*, 2008; Urban & Vogel, 2008). Nonetheless, GlmY carries all elements required to bind RapZ. Through molecular mimicry, GlmY is able to sequester RapZ, leaving GlmZ unprocessed (Göpel *et al*, 2013, 2016). Sponging of protein or sRNA by decoy RNAs has emerged as a widespread principle in bacterial post-transcriptional regulation (Sonnleitner & Bläsi, 2014; Miyakoshi *et al*, 2015; Romeo & Babitzke, 2018). GlmY specifically accumulates and counters GlmZ decay, when GlcN6P levels decrease (Reichenbach *et al*, 2008; Khan *et al*, 2016). Accordingly, GlmS amounts increase and GlcN6P is replenished. Ultimately, this mechanism achieves GlmS feedback regulation, ensuring GlcN6P homeostasis and thereby cell envelope synthesis. GlcN6P is the source of all amino sugar containing constituents of the cell wall and also of the outer membrane of Gram-negative bacteria. Bacteria can procure GlcN6P from external amino sugars such as glucosamine (GlcN), which can be taken up and converted to GlcN6P (Alvarez-Anorve *et al*, 2016). If not available, GlcN6P must be synthesized by GlmS (Milewski, 2002). GlmS is also target for antibiotics produced by other microorganisms and GlmY/GlmZ provide protection as they overcome inhibition by increasing GlmS amounts (Khan *et al*, 2016), a defense that could not be achieved by allosteric regulation of the enzyme. Hence, the need for GlcN6P synthesis may strongly vary during the bacterial life cycle and GlmS activity needs tight and instant control. To this end, cells must measure intracellular GlcN6P, but how this is achieved in *E. coli* is unknown so far.

Whereas transcription of *glmZ* is constitutive in *E. coli*, *glmY* expression is tightly controlled (Göpel *et al*, 2011). GlmY can be transcribed from two overlapping $\sigma^{54}$ and $\sigma^{70}$ promoters, albeit the weaker $\sigma^{70}$ promoter is usually repressed by binding of $\sigma^{54}$ to the overlapping $-24/-12$ sequence motif (Urban *et al*, 2007; Reichenbach *et al*, 2009). The stronger $\sigma^{54}$ promoter is controlled by the two-component system (TCS) QseE/QseF (a.k.a. GlrK/GlrR or YfhK/YfhA) consisting of histidine kinase QseE and response regulator QseF, which activates *glmY* transcription when phosphorylated (Göpel *et al*, 2011). QseE/QseF employ the third component QseG, which is essential for activity of this TCS (Göpel & Görke, 2018). QseG is a lipoprotein attached to the inner leaflet of the outer membrane and activates kinase QseE by interaction in the periplasm.

So far, it remained mysterious how the GlcN6P signal is sensed and processed by the GlmY/RapZ/GlmZ system. As it appeared to act upstream, GlmY was a likely candidate. However, in the current study, we identify protein RapZ as a *bona fide* receptor for GlcN6P. RapZ binds this metabolite and is required for the GlcN6P response *in vivo*. Upon metabolite depletion, RapZ activates QseE/QseF to upregulate expression of its decoy GlmY, which subsequently sequesters RapZ into stable complexes, titrating it away from GlmZ. When GlcN6P is replenished, GlmY is released and rapidly degraded. Thus, RapZ regulates and is regulated by sRNA GlmY in response to metabolite availability. We unveil RapZ as a multifunctional RBP that not only targets an sRNA to degradation by RNase E, but also acts as sensor communicating the cellular GlcN6P status to a TCS to adjust expression of its titrating decoy.

## Results

### The small RNA GlmY/GlmZ circuit requires protein RapZ for sensing GlcN6P

The degree of sRNA GlmZ processing is determined by availability of adaptor protein RapZ for interaction, which is in turn regulated by sRNA GlmY. GlmY levels rise when GlcN6P concentrations decrease. Previous work showed that a *glmY* mutant fails to stabilize full-length GlmZ and to upregulate GlmS synthesis in response to GlcN6P scarcity (Reichenbach *et al*, 2008; Khan *et al*, 2016). Therefore, we initially hypothesized that GlmY or a factor upstream might sense GlcN6P, e.g., through a riboswitch mechanism as observed in Gram-positive bacteria (Collins *et al*, 2007). If so, GlmY should still respond to GlcN6P in a strain lacking RapZ.

To test this, we compared levels of GlmY and also GlmZ in *wild-type* and *ΔrapZ* strains under conditions of GlcN6P sufficiency and depletion. To monitor the regulatory output, the strains carried an ectopic *glmS'-lacZ* reporter fusion in the chromosome. To trigger GlcN6P depletion, we used Nva-FMDP, a synthetic derivative of an antibiotic, which selectively inhibits GlmS enzymatic activity and causes exhaustion of intracellular GlcN6P (Chmara *et al*, 1998). We previously demonstrated that Nva-FMDP upregulates *glmS* expression in a concentration-dependent manner through activation of the GlmY/GlmZ system and the presence of an exogenous amino sugar overrides this effect (Khan *et al*, 2016). Cultures grown to exponential phase were split, and sub-cultures were provided with a sub-inhibitory concentration of Nva-FMDP or $H_2O$ as mock control. Growth was continued, and samples were harvested hourly for Northern analysis of total RNA and determination of β-galactosidase activities (Fig 1A; Appendix Fig S1). In the *wild-type* strain, Nva-FMDP caused accumulation of processed GlmY, which concomitantly inhibited processing of GlmZ leading to increased *glmS'-lacZ* expression, as expected. Previous analysis already showed that also GlmY undergoes processing in its 3′ end converting the 184 nt long primary sRNA to a ~148 nt long 5′ cleavage product (subsequently designated GlmY*; Vogel *et al*, 2003), which accumulates *in vivo* and sequesters RapZ (Reichenbach *et al*, 2008; Göpel *et al*, 2013).

In the *ΔrapZ* mutant, processing of GlmZ was abolished, reflecting the requirement of RapZ for cleavage of GlmZ by RNase E (Göpel *et al*, 2013, 2016). As expected, accumulation of full-length GlmZ caused high *glmS'-lacZ* expression, regardless of the GlcN6P level (Fig 1A, bottom). Importantly, Nva-FMDP did not trigger accumulation of GlmY* in the *ΔrapZ* mutant. Moreover, in agreement with previous results (Reichenbach *et al*, 2008), GlmY levels were collectively decreased in the *ΔrapZ* mutant. To verify these results for a wider range of *E. coli* K-12 strains, we repeated the experiment, which was performed using derivatives of CSH50 (Miller, 1972), also in derivatives of MG1655 (Blattner *et al*, 1997). Comparable results were obtained, confirming that RapZ is required for accumulation of GlmY* in response to GlcN6P depletion and that GlmY cannot sense GlcN6P on its own (Appendix Fig S2).

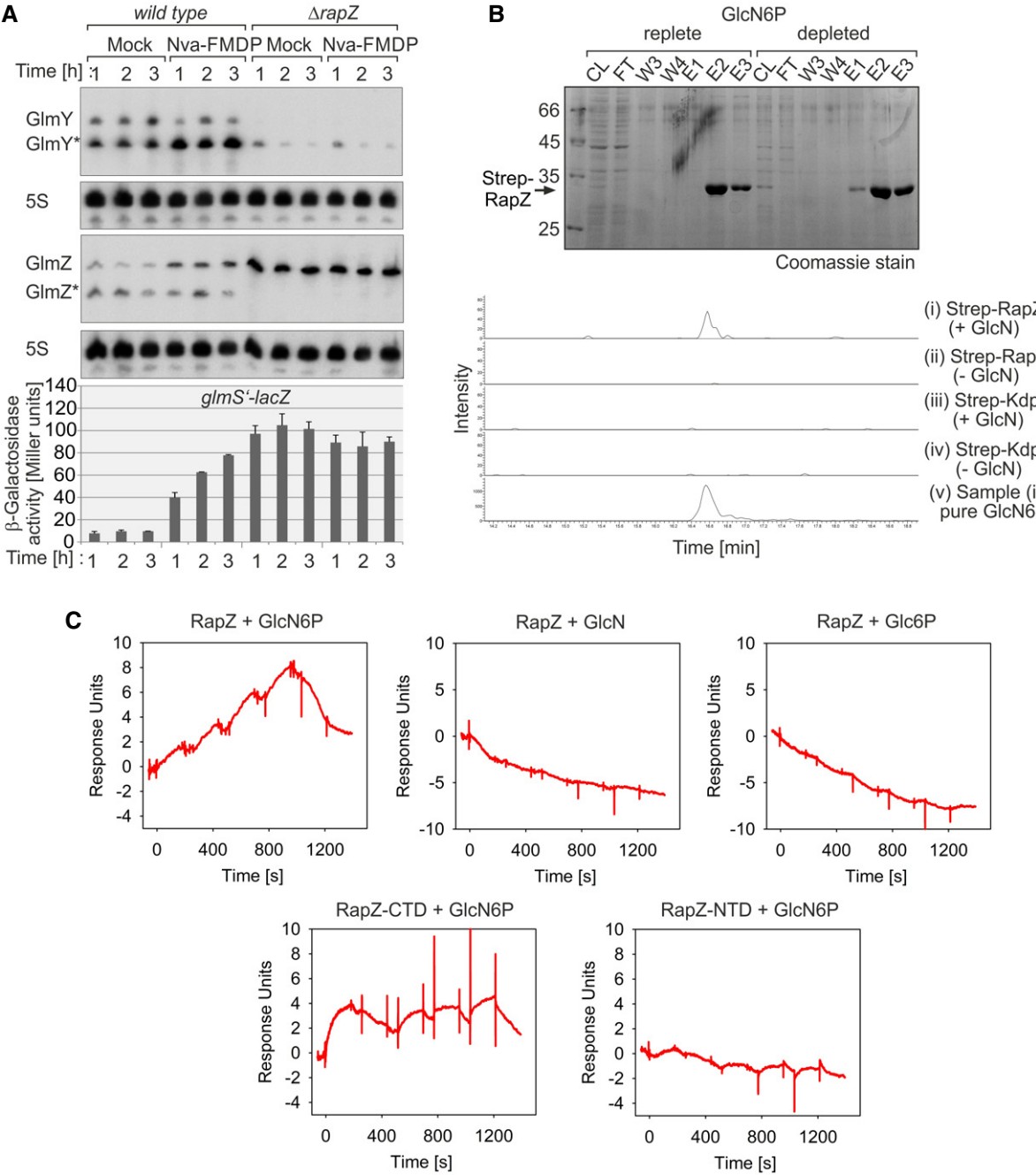

**Figure 1. RapZ binds GlcN6P and is required for regulation of the GlmY/GlmZ/*glmS* cascade by GlcN6P.**

A Northern blots comparing GlmY and GlmZ levels in *wild-type* strain Z8 and the Δ*rapZ* mutant Z28 under normal growth and GlcN6P starvation conditions. Both strains, which also carried a chromosomal *glmS'-lacZ* fusion, were treated with 60 μg/ml Nva-FMDP or H₂O ("mock"). Samples were harvested hourly for Northern analysis and determination of β-galactosidase activity. Growth curves are shown in Appendix Fig S1. Blots were re-probed using a 5S rRNA specific probe to provide loading controls.

B The purification profile of Strep-RapZ from the *ΔglmS* strain Z904 under GlcN6P replete and depletion conditions is shown (top). The cleared lysate (CL), flow through (FT), washing steps (W), and the elution fractions (E1-3) from Strep-Tactin affinity chromatography were separated on 12.5% SDS-PAA gels and stained with Coomassie blue. Metabolites were extracted from E2 and analyzed by HILIC-MS/MS. The extracted ion chromatograms of the LC-MS analysis targeting GlcN6P (retention time 16.6 min) are shown below. The samples derived from purification of Strep-RapZ (panels i and ii) or Strep-KdpE (panels iii and iv) were analyzed with the SRM transition m/z 258.1 to m/z 97 in the negative ion mode. The identity of the metabolite detected in panel i was confirmed by adding chemically pure GlcN6P to a final concentration of 100 pg/μl (panel v).

C SPR analysis addressing interaction of RapZ variants with GlcN6P and similar metabolites. The Strep-tagged proteins were captured onto a sensor chip, and various concentrations of the respective metabolite (i.e., 100, 500, 1,000, 2,500, and 5,000 nM) were injected using a single-cycle kinetics approach.

Data information: In (A), β-galactosidase activities are presented as mean ± SD. *n* = 2.
Source data are available online for this figure.

## RapZ binds GlcN6P

Our results (Fig 1A) suggested that RapZ acts upstream of GlmY with respect to GlcN6P sensing. One possibility is that RapZ senses GlcN6P, perhaps by binding, and accordingly modulates GlmY amounts. To determine whether RapZ binds GlcN6P, we tested whether GlcN6P co-elutes upon purification of Strep-tagged RapZ from cells grown under GlcN6P replete conditions by affinity chromatography (Fig 1B). As a control, we purified Strep-RapZ from GlcN6P depleted cells that were obtained by shifting Δ*glmS* cells grown in the presence of GlcN to a medium devoid of amino sugars. Metabolites were extracted from protein elution fractions, and a targeted metabolomics approach was employed for identification of GlcN6P. Fig 1B shows the extracted ion chromatogram of the LC-MS analysis targeting GlcN6P (retention time 16.6 min). Panel i displays the analysis of the RapZ sample obtained under GlcN6P replete conditions. The identity of the detected metabolite was confirmed by analyzing three selected reaction monitoring (SRM) transitions (Appendix Fig S3) and by adding chemically pure GlcN6P to the Strep-RapZ sample obtained under GlcN6P replete conditions (Fig 1B, panel v). No GlcN6P signal above noise was detected when Strep-RapZ was purified under GlcN6P depletion conditions (Fig 1B, panel ii). Likewise, GlcN6P was undetectable when the unrelated protein KdpE (Heermann *et al*, 2009) was purified under the same conditions (Fig 1B, panels iii, iv; Appendix Fig S4). These data indicate that Strep-RapZ interacts with GlcN6P *in vivo*.

We used surface plasmon resonance (SPR) spectroscopy to study interaction of RapZ with the metabolite *in vitro*. Purified Strep-RapZ was captured onto a sensor chip, and increasing concentrations of GlcN6P were injected. Interaction became detectable and calculations revealed an overall affinity ($K_D$) of 186 nM for GlcN6P ($k_a = 2.4 \times 10^4$ M$^{-1}$ s$^{-1}$, $k_d = 4.4 \times 10^{-3}$ s$^{-1}$; Fig 1C top left panel). No response was observed when using structurally related metabolites such as GlcN or glucose-6-phosphate (Glc6P) indicating that interaction of RapZ with GlcN6P is highly specific (Fig 1C top panels).

## RapZ upregulates *glmY* transcription in response to GlcN6P depletion

RapZ binds GlcN6P and triggers accumulation of its decoy GlmY under conditions of GlcN6P scarcity (Fig 1; Appendix Fig S2). This upregulation could take place at the transcriptional or post-transcriptional level, or at both. To explore the mode of regulation, we first studied the role of GlcN6P for expression of *glmY*. We subjected a strain carrying an ectopic *glmY'-lacZ* reporter fusion in the chromosome to various degrees of GlcN6P depletion. An exponentially growing culture was split, and sub-cultures in 96-well plates were supplied with various sub-inhibitory concentrations of Nva-FMDP. Subsequently, β-galactosidase activities were recorded at hourly time intervals. Interestingly, Nva-FMDP caused upregulation of *glmY* expression in a concentration-dependent manner (Fig 2A; Appendix Fig S5 for MG1655 derivatives). A *glmZ'-lacZ* fusion, which was included for comparison, did not respond. Notably, Nva-FMDP had no effect on *glmY* expression in the Δ*rapZ* mutant (Fig 2B). Moreover, the Δ*rapZ* mutation reduced *glmY* expression levels even before Nva-FMDP was added (i.e., at *t* = 0). We further assessed the requirement of RapZ for *glmY* promoter activity using

cultures grown in flasks under standard conditions (i.e., without eliciting GlcN6P starvation). In this case, *glmY* promoter activity dropped fourfold in the Δ*rapZ* mutant (Fig 2E). Complementation with plasmid borne *rapZ* restored *glmY* expression above *wild-type* levels (Fig 2E).

To determine which of the two promoters known to drive *glmY* transcription is regulated by GlcN6P, we tested strains carrying *glmY'-lacZ* fusions comprising mutations that selectively inactivate either the σ$^{70}$ or the σ$^{54}$ promoter (Reichenbach *et al*, 2009). The reporter assays revealed that Nva-FMDP increases *glmY* expression exclusively through the σ$^{54}$ promoter (Fig 2C). Introduction of a Δ*qseF* mutation abolished expression from the σ$^{54}$ promoter (Fig 2D), reflecting the absolute requirement of σ$^{54}$ promoters for their cognate enhancer binding proteins (Bush & Dixon, 2012). Importantly, Nva-FMDP was unable to increase *glmY* expression in this strain. Taken together, RapZ stimulates the σ$^{54}$ promoter of *glmY* in a QseF-dependent manner and increases expression further when the intracellular GlcN6P concentration drops.

## RapZ interacts with QseE and QseF

RapZ is a RBP making a direct interaction with the *glmY* promoter region unlikely. Hence, modulation of QseEGF or of σ$^{54}$ activity by RapZ appeared to be feasible mechanisms. The latter possibility is reinforced by the conserved co-localization of *rpoN* (encoding σ$^{54}$) and *rapZ* in one operon. However, the absence of *rapZ* had no significant impact on transcription of other σ$^{54}$-dependent genes, as judged from reporter assays using *lacZ* fusions to the promoters of *glnA* and *zraP* (Appendix Fig S6A and B), both of which are controlled by σ$^{54}$ (Bonocora *et al*, 2015). This observation made a global effect of RapZ on σ$^{54}$ activity unlikely.

Alternatively, we considered modulation of QseE or QseF activity by RapZ. As several TCSs are regulated by interaction with accessory proteins (Buelow & Raivio, 2010), we tested whether RapZ binds QseE and/or QseF, using the bacterial adenylate cyclase-based two-hybrid (BACTH) assay (Karimova *et al*, 1998). Indeed, enzyme assays indicated that RapZ interacts with both QseE and QseF (Fig 3A). Interaction was also detectable when protein fusion partners were swapped and irrespective of whether an MG1655::*cyaA* derivative or the original BACTH reporter strain was used (Fig 3A; Appendix Fig S7A). Interaction persisted in a derivative strain lacking the endogenous *qseEGF* operon suggesting that RapZ binds both proteins independent of each other (Appendix Fig S7B).

For confirmation, interaction of RapZ with QseE and QseF was tested using SPR spectroscopy. N-terminally Strep-tagged RapZ, which retains the ability to activate *glmY* expression (Fig 2E), was captured onto the sensor chip before incremental concentrations of the analytes QseE-His$_{10}$ or QseF-His$_{10}$ were injected (Fig 3B; Appendix Fig S8). In case of QseE, the soluble cytoplasmic C-terminal part comprising the HAMP and the transmitter domains (subsequently designated QseE') was used. Interaction of RapZ with both QseE' and QseF could be observed (Fig 3B). Quantification revealed overall affinities ($K_D$) of 12 μM for the QseE'/RapZ interaction ($k_a = 4.5 \times 10^3$ M$^{-1}$ s$^{-1}$; $k_d = 5.4 \times 10^{-2}$ s$^{-1}$) and 42 μM for the QseF/RapZ interaction ($k_a = 9.8 \times 10^2$ M$^{-1}$ s$^{-1}$; $k_d = 4.1 \times 10^{-3}$ s$^{-1}$). However, a clear 1:1 interaction is not represented by the sensorgrams, as no saturation in binding was obtained at high analyte concentrations, presumably due to formation of

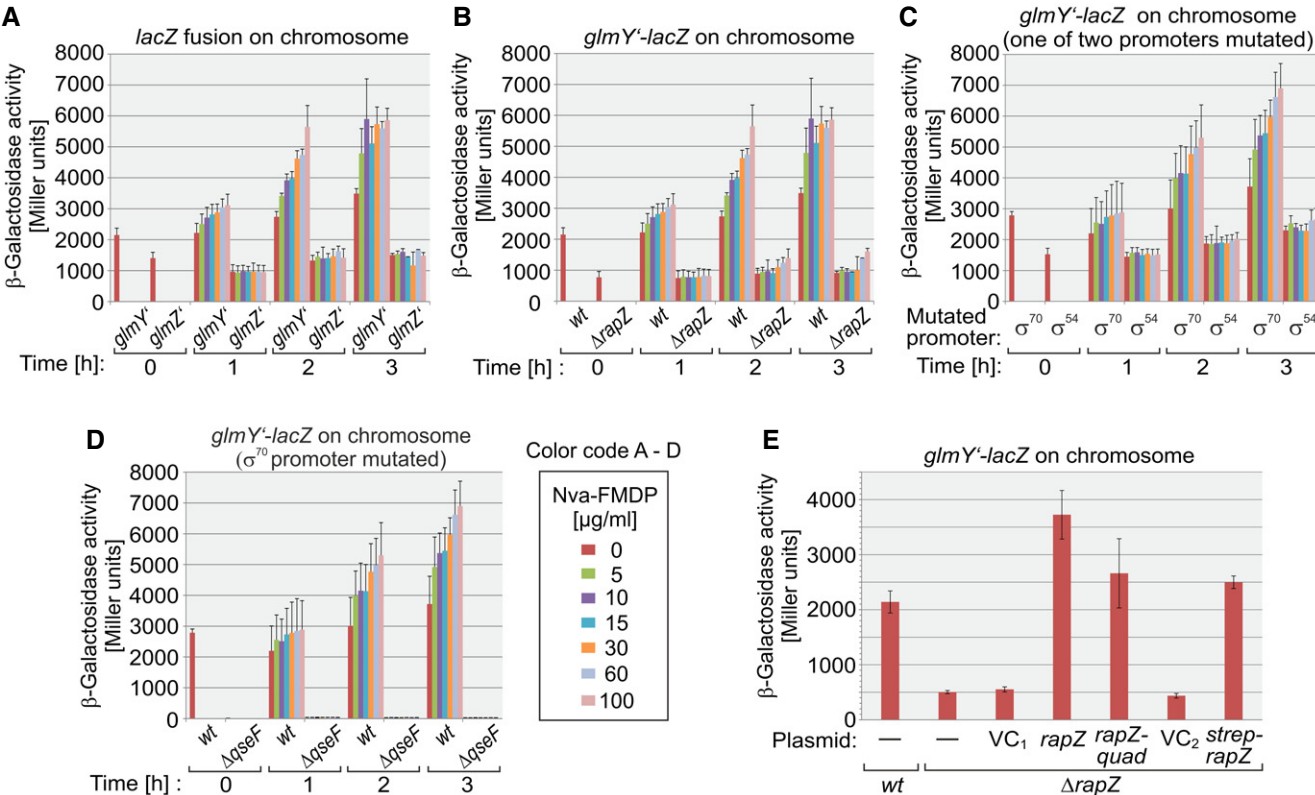

**Figure 2.   RapZ is required for full activity of the *glmY* σ⁵⁴ promoter and upregulates *glmY* expression under GlcN6P starvation conditions.**

Reporter gene assays addressing expression of *lacZ* fusions under GlcN6P replete and depletion conditions. In (A) to (D), strains were grown in 96-well plates and exposed to various degrees of GlcN6P depletion elicited by Nva-FMDP. Cells were harvested at indicated times, and the β-galactosidase activities were determined.

A   Strains Z197 and Z360 were used, which harbor *glmY'-lacZ* and *glmZ'-lacZ* fusions, respectively.

B   Expression of *glmY'-lacZ* in strain Z197 and the *ΔrapZ* mutant Z225 is compared.

C   Strains Z190 and Z201 were addressed, which transcribe *glmY'-lacZ* from either the σ⁵⁴ promoter or the σ⁷⁰ promoter, respectively.

D   Strains Z190 and the *ΔqseF* mutant Z196 are compared, both of which transcribe the *glmY'-lacZ* fusion solely from the σ⁵⁴ promoter.

E   Complementation experiment analyzing the requirement of *rapZ* for *glmY* expression in cells grown to exponential phase under standard conditions in flask cultures. Strains Z197 and the *ΔrapZ* mutant Z225 were used. Tested plasmids were pFDX4291 (vector control for pFDX4324 and pYG82 = VC₁), pFDX4324 (*rapZ*), pYG82 (*rapZ_quad*), pBGG237 (vector control for pBGG164 = VC₂), and pBGG164 (*strep-rapZ*).

Data information: Note that experiments (A) and (B) as well as (C) and (D) were performed in parallel, respectively. Therefore, the same values are presented for strains Z197 and Z190. In (A–E), β-galactosidase activities are presented as mean ± SD. (A–D): $n = 3$; (E): $n = 4$.

Source data are available online for this figure.

aggregates. Overall, we conclude that RapZ interacts with both QseE and QseF, but transiently as indicated by the high dissociation rates observed by SPR.

RapZ consists of well separated N- and C-terminal globular domains (NTD and CTD), which form homodimers on their own (Gonzalez *et al*, 2017). BACTH analysis reveals that the separately synthesized domains of RapZ are incapable of interacting efficiently with QseF or QseE (Fig EV1A). Although the RapZ-CTD retains some interaction, this residual binding is not sufficient to activate *glmY* expression as shown by a complementation assay using low copy plasmids encoding the RapZ variants (Fig EV1B). Moreover, introduction of an Asp182Ala substitution into full-length RapZ abrogating self-interaction of the CTD (Gonzalez *et al*, 2017), concomitantly abolishes interaction with QseE as well as QseF (Fig EV1A). Apparently, both domains of RapZ contribute to binding and activation of QseE/QseF and proper oligomerization of RapZ is a prerequisite for interaction.

**RapZ stimulates phosphorylation of the QseE/QseF TCS**

Accessory proteins frequently influence activity of TCSs by modulation of their phosphorylation state (Buelow & Raivio, 2010). To determine whether RapZ impacts phosphorylation of response regulator QseF, we compared plasmid-encoded QseF variants carrying exchanges in the Asp56 phosphorylation site in the receiver domain. That is, we monitored the activities of *wild-type* QseF and non-phosphorylatable QseF variants in *ΔrapZ* and *rapZ⁺* strains by using the *glmY'-lacZ* fusion as reporter (Fig 3C). To avoid interference with the σ⁷⁰ promoter, the *glmY'-lacZ* fusion was solely driven from the σ⁵⁴ promoter (i.e., the σ⁷⁰ promoter was mutated). The presence of the empty vector (VC) resulted in very low β-galactosidase activities during growth, reflecting the requirement of the *glmY* σ⁵⁴ promoter for QseF (Fig 3C). Complementation of the *rapZ⁺* strain with the plasmid encoding *wild-type* QseF resulted in intermediate *glmY'-lacZ* levels, whereas ~2-fold lower activities were measured in the

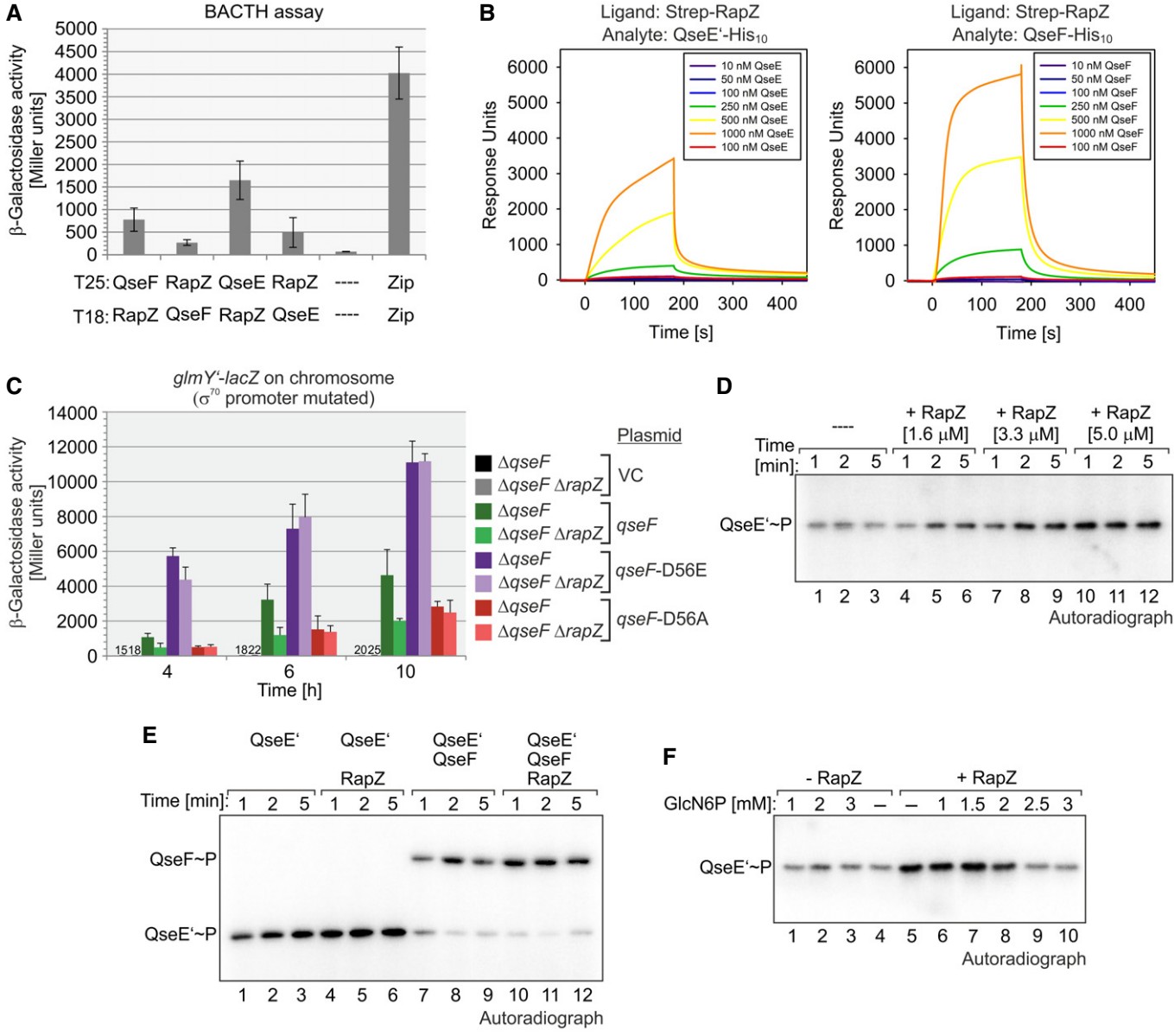

**Figure 3. RapZ stimulates phosphorylation of the TCS QseE/QseF by interaction and GlcN6P counters this process.**

A  BACTH assay addressing interaction of RapZ with QseF and QseE. The following plasmid combinations were tested using reporter strain RH785 (columns 1–6): pBGG352/pBGG349, pBGG353/pBGG348, pYG199/pBGG349, pYG246/pBGG348, pKT25/pUT18C (negative control), and pKT25-zip/pUT18C-zip (positive control).

B  SPR spectroscopy experiments addressing interaction of RapZ with the cytoplasmic part (aa 196–475) of kinase QseE and response regulator QseF. Strep-RapZ was captured onto a sensor chip, and increasing concentrations of QseE'-His$_{10}$ or QseF-His$_{10}$ were injected.

C  To assess the role of RapZ for QseF activity, β-galactosidase activities were determined from strains Z196 (ΔqseF) and Z1110 (ΔqseF ΔrapZ) at indicated times during growth. Strains harbored the glmY'-lacZ fusion that is solely expressed from the σ$^{54}$ promoter and the following plasmids: pKESK23 (VC = vector control; black and gray; note that activities are too low for display), pYG89 (qseF, green), pYG90 (qseF-D56E, purple), and pYG93 (qseF-D56A, red).

D  In vitro phosphorylation assays addressing autophosphorylation of 1 μM His$_{10}$-tagged QseE' (aa 196–475) in the presence of various concentrations of RapZ. Samples were removed following [γ-$^{32}$P]-ATP addition at indicated times and separated on 12.5% SDS-PAA gels, which were analyzed by phospho-imaging.

E  Analysis of QseE' autophosphorylation (lanes 1–6) and phosphoryl-group transfer to QseF (lanes 7–12) in the absence or presence of 5 μM Strep-RapZ. To assess phosphoryl-group transfer, 1 μM QseF-His$_{10}$ was added to the assay.

F  To analyze the role of GlcN6P, 5 μM RapZ or the equivalent volume of buffer was pre-incubated with the indicated GlcN6P concentration for 5 min and subsequently 1 μM QseE'-His$_{10}$ was added. Following an additional incubation for 5 min, [γ-$^{32}$P]-ATP was added and the reactions were stopped after 1 min.

Data information: In (A) and (C), β-galactosidase activities are presented as mean ± SD. (A): $n = 3$; (C): $n = 4$.
Source data are available online for this figure.

presence of the QseF-D56A variant, which mimics non-phosphory-lated QseF (Fig 3C). The latter activities reflect the residual ability of non-phosphorylated QseF to activate *glmY* expression when over-produced (Göpel & Görke, 2018). High *glmY* expression levels were obtained in the presence of the QseF-D56E variant, which mimics phosphorylated QseF (Fig 3C). Importantly, the *glmY* expression levels caused by the QseF-D56A and QseF-D56E variants remained unaffected by the *ΔrapZ* mutation. In contrast, the *glmY* expression level triggered by *wild-type* QseF dropped ~2.5-fold in the *ΔrapZ* mutant, i.e., roughly to the level observed for QseF-D56A (Fig 3C). Hence, RapZ stimulates activity of *wild-type* QseF, but not of non-phosphorylatable QseF mutants.

To obtain further insight, we performed *in vitro* phosphorylation assays using purified recombinant proteins and [γ-$^{32}$P] ATP. Aliquots were withdrawn, and reactions were stopped at indicated times to follow protein phosphorylation over time. When incubated alone, a phosphorylation signal for QseE' became detectable reflecting autophosphorylation activity (Fig 3D). Intriguingly, the QseE' phosphorylation signal increased concomitantly with incre-mental concentrations of RapZ. The strongest QseE' phosphoryla-tion signal was obtained when RapZ was in 5 × molar excess over QseE' (Fig 3D, lanes 10–12; Fig 3E, lanes 4–6). When QseE' and QseF were co-incubated, the signal for QseE'~P strongly decreased and phosphorylated QseF became visible, reflecting phosphoryl-group transfer (Fig 3E, lanes 7–9). Importantly, the presence of RapZ increased phosphorylation of QseF (Fig 3E, lanes 10–12). Our genetic analyses suggested that GlcN6P-free RapZ activates QseE/QseF (Figs 1 and 2). Indeed, the presence of 2 mM GlcN6P inhibited and higher concentrations abolished stimulation of QseE' autophos-phorylation by RapZ (Fig 3F, lanes 5–10). GlcN6P alone had no effect on QseE' (Fig 3F, lanes 1–4). Taken together, GlcN6P-free RapZ stimulates autophosphorylation of QseE by interaction, result-ing in increased phosphorylation and thereby activity of response regulator QseF.

## Small RNAs GlmY and GlmZ counteract activation of QseE/QseF by RapZ

Under GlcN6P depletion conditions, RapZ is licensed to activate QseE/QseF (Figs 2 and 3). We wondered how this response is limited to attenuate the burst of GlmY production. As GlmY seques-ters RapZ, one possibility is that GlmY itself, i.e., sRNA binding, might prevent RapZ from ongoing activation of QseE/QseF.

Interestingly, a double mutant lacking both sRNAs constantly produced somewhat higher *glmY'-lacZ* expression levels during growth when compared to the *wild-type*, suggesting that the sRNAs repress *glmY* transcription to some extent (Fig 4A). We performed the complementary experiment and overexpressed *glmY* and *glmZ* from plasmids in *rapZ*[+] and *ΔrapZ* strains, respectively. In the pres-ence of the empty vector (VC), ~ 5-fold lower *glmY'-lacZ* expression levels were detected in the *ΔrapZ* mutant as compared to the *rapZ*[+] strain (Fig 4B), recapitulating that RapZ is also required for undis-turbed *glmY* expression under standard growth conditions (cf. Fig 2E). Interestingly, overexpression of GlmY or GlmZ in the *rapZ*[+] strain reduced *glmY* expression to the level observed in the *ΔrapZ* mutant, whereas overexpression of the unrelated sRNA GcvB had no effect (Fig 4B). Moreover, the low *glmY* expression level in the *ΔrapZ* mutant remained unaffected by overproduction of the

sRNAs. As deletion or overproduction of GlmY or GlmZ has no impact on RapZ levels (Durica-Mitic & Görke, 2019), these results suggested that GlmY and GlmZ counteract activation of QseE/QseF by RapZ.

For confirmation, we tested autophosphorylation of kinase QseE' *in vitro* in the presence of RapZ and various concentrations of the *in vitro* transcribed sRNAs (i.e., 0.5, 1.25, 2.5 μM; Appendix Fig S9). As observed before, the presence of 5 μM RapZ stimulated QseE' phosphorylation (Fig 4C, lanes 1 and 2). Intriguingly, the additional presence of at least 1.25 μM GlmY* or GlmZ decreased the QseE' autophosphorylation signal roughly to the intensity observed in the absence of RapZ, whereas sRNA GcvB was without effect (Fig 4C, lanes 3–11). Control assays ruled out that GlmY* or GlmZ has a direct effect on QseE' autophosphorylation (Fig 4C, lanes 12–16). These results show that sRNAs GlmY* and GlmZ directly counteract activation of QseE autophosphorylation by RapZ. Hence, GlmY* is capable of limiting its own production by sequestration of RapZ, providing a negative feedback loop.

## GlcN6P starvation increases GlmY* half-life dramatically

RapZ increases expression of its decoy sRNA GlmY* when GlcN6P is limiting (Figs 1 and 2). Assuming this mechanism as the only regulatory layer, QseE/QseF should be essential for the response to GlcN6P depletion. However, previous work has shown that GlmY* accumulates upon GlcN6P scarcity to a limited extent even in *ΔqseE* and *ΔqseF* mutants (Reichenbach *et al*, 2009). To verify these results, which were obtained using derivatives of strain CSH50, we repeated the experiment with MG1655 derivatives. Cultures of the *wild-type* and the *qseE* and *qseF* mutant strains were subjected to GlcN6P sufficiency and starvation conditions using Nva-FMDP. As expected, much lower GlmY* amounts were detectable in the mock-treated cultures of the *ΔqseF* and *ΔqseE* mutants as compared to the *wild-type*, reflecting that *glmY* expression is solely driven from the σ$^{70}$ promoter (Fig EV2). Nonetheless, addition of Nva-FMDP still triggered residual upregulation of GlmY* in the mutants, confirming the previous findings. As the σ$^{70}$ promoter of *glmY* does not respond to GlcN6P starvation (Fig 2C), this remaining increase of GlmY amounts can only be explained by a post-transcriptional mecha-nism.

Hence, we determined the half-life of GlmY* under conditions of GlcN6P sufficiency and depletion. The *wild-type* strain (CSH50 derivative) was grown in the presence or absence of Nva-FMDP, and rifampicin was added to stop transcription. Samples were harvested at various times for Northern blot analysis. Both GlmY* and full-length GlmZ were short-lived in the mock-treated cells with half-lives of ≤ 3 min, respectively (Fig 5A). Interestingly, half-life of GlmY* increased to ~13 min upon Nva-FMDP treatment (Fig 5A and C), increasing half-life of full-length GlmZ concomitantly. We obtained comparable results when using a MG1655 derivative strain (Appendix Fig S10A and C).

Next, we determined whether stabilization of GlmY* is rever-sible, i.e., abolished when GlcN6P becomes available again follow-ing a period of GlcN6P starvation. In this case, we cultivated a *ΔglmS* mutant in medium devoid of amino sugars to elicit progres-sive GlcN6P depletion, which ultimately leads to cell lysis (Fig EV3). Concomitantly, steady-state levels of GlmY* increase, counteracting processing of GlmZ, which in turn activates *glmS*

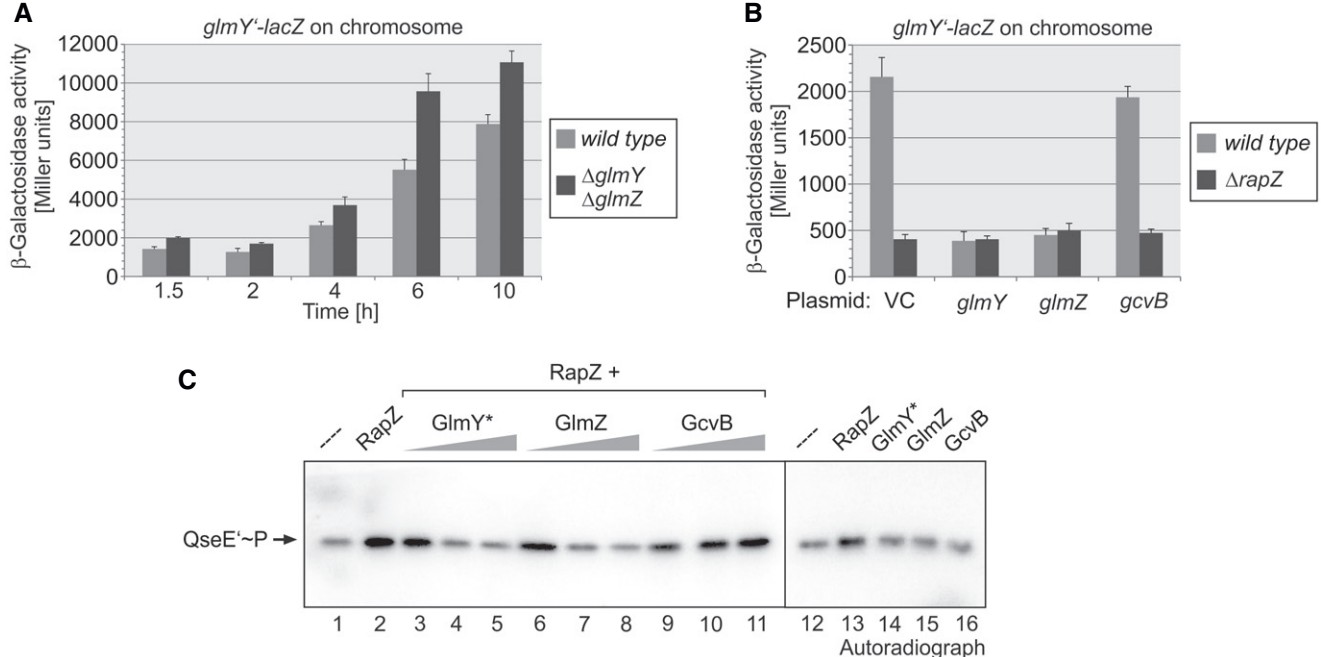

**Figure 4. sRNAs GlmY and GlmZ counteract activation of QseE/QseF by RapZ.**

A β-Galactosidase activities of strains Z197 (*wild-type*) and Z1118 (*ΔglmY ΔglmZ*), which carry the chromosomal *glmY'-lacZ* fusion, were determined during growth.

B Strains Z197 (*wild-type*) and Z225 (*ΔrapZ*) were transformed with the following plasmids expressing the mentioned sRNAs: pBR-plac (vector control = VC), pYG83 (*glmY*), pYG84 (*glmZ*), and pSD69 (*gcvB*). sRNA expression was induced with 1 mM IPTG, and β-galactosidase activities were determined in the exponential growth phase.

C To assess the impact of GlmY* and GlmZ on stimulation of QseE' autophosphorylation by RapZ, 1 μM QseE'-His$_{10}$ was incubated with [γ-$^{32}$P]-ATP in the absence or presence of 5 μM Strep-RapZ and/or the sRNAs GlmY*, GlmZ and GcvB. In lanes 3–11, QseE' was co-incubated with RapZ as well as the indicated sRNAs provided at 0.5, 1.25, and 2.5 μM. In lanes 14–16, QseE' was incubated with 2.5 μM of each sRNA without RapZ. Samples were removed 1 min after addition of [γ-$^{32}$P]-ATP and separated on 12.5% SDS-PAA gels, which were analyzed by phospho-imaging.

Data information: In (A) and (B), β-galactosidase activities are presented as mean ± SD. (A): $n = 4$; (B): $n = 3$.
Source data are available online for this figure.

expression. For half-life determination, the *ΔglmS* mutant was grown in the absence or presence of GlcN before transcription was stopped. In the GlcN6P replete cells, both GlmY* and GlmZ became rapidly degraded exhibiting half-lives < 3 min (Fig 6A and C; Appendix Fig S11). In agreement with the measurements using Nva-FMDP, half-life of GlmY* dramatically increased in the GlcN6P depleted cells ($t_{1/2}$ > 30 min), concomitantly inhibiting processing of GlmZ. To test whether GlcN6P destabilizes GlmY*, the culture grown in the absence of GlcN was split at $t = 8$ min following rifampicin addition, and one of the sub-cultures was resupplied with GlcN (arrow in Fig 6A). Replenishment of GlcN6P rapidly destabilized GlmY*, whereas it remained stable in the culture lacking GlcN (Fig 6A; Appendix Fig S12A and D for MG1655 *ΔglmS* derivative). The role of GlcN6P for GlmY* stability is independent of GlmZ, as GlcN6P depletion provoked GlmY* stabilization also in a *ΔglmS ΔglmZ* double mutant (Appendix Fig S13A). On the other hand, GlcN6P depletion did not increase half-life of GlmZ in a *ΔglmY ΔglmS* mutant strain (Appendix Fig S13B), confirming that GlmY is essential for this response. Overall, these data show that GlcN6P availability destabilizes sRNA GlmY*. Thus, accumulation of GlmY* in response to GlcN6P depletion (Fig 1A) is the consequence of both, its higher expression and increased stability.

## Interaction with RapZ protects GlmY from degradation under GlcN6P starvation

As RapZ is sequestered by GlmY* under GlcN6P depletion conditions (Göpel *et al*, 2013), it is the likely candidate responsible for stabilization of the sRNA. Indeed, introduction of a *ΔrapZ* mutation abolished the residual increase of steady-state GlmY* levels observed in the *ΔqseF* mutant under GlcN6P limitation, supporting this idea (Fig EV2, bottom). Consequently, we assessed GlmY* and also GlmZ stability in the *ΔrapZ* mutant. Full-length GlmZ was stabilized in this strain regardless of the absence or presence of Nva-FMDP, as expected (Fig 5B and C). In contrast, GlmY* was short-lived exhibiting comparable half-lives ($t_{1/2}$ < 2 min) under both conditions (Fig 5B and C; Appendix Fig S10B and C for MG1655 derivatives). Stabilization of GlmY* was also not observed, when a *ΔglmS* mutation was used to elicit GlcN6P depletion in the *ΔrapZ* mutant (Fig 6B and C; Appendix Fig S12B and C for MG1655 derivative).

Thus, RapZ protects GlmY* from degradation when GlcN6P is limiting, most likely through binding. To address this issue, we used a RapZ mutant (RapZ$_{quad}$) that carries a quadruple exchange in the CTD, abolishing RNA-binding activity (Göpel *et al*, 2013). Notably,

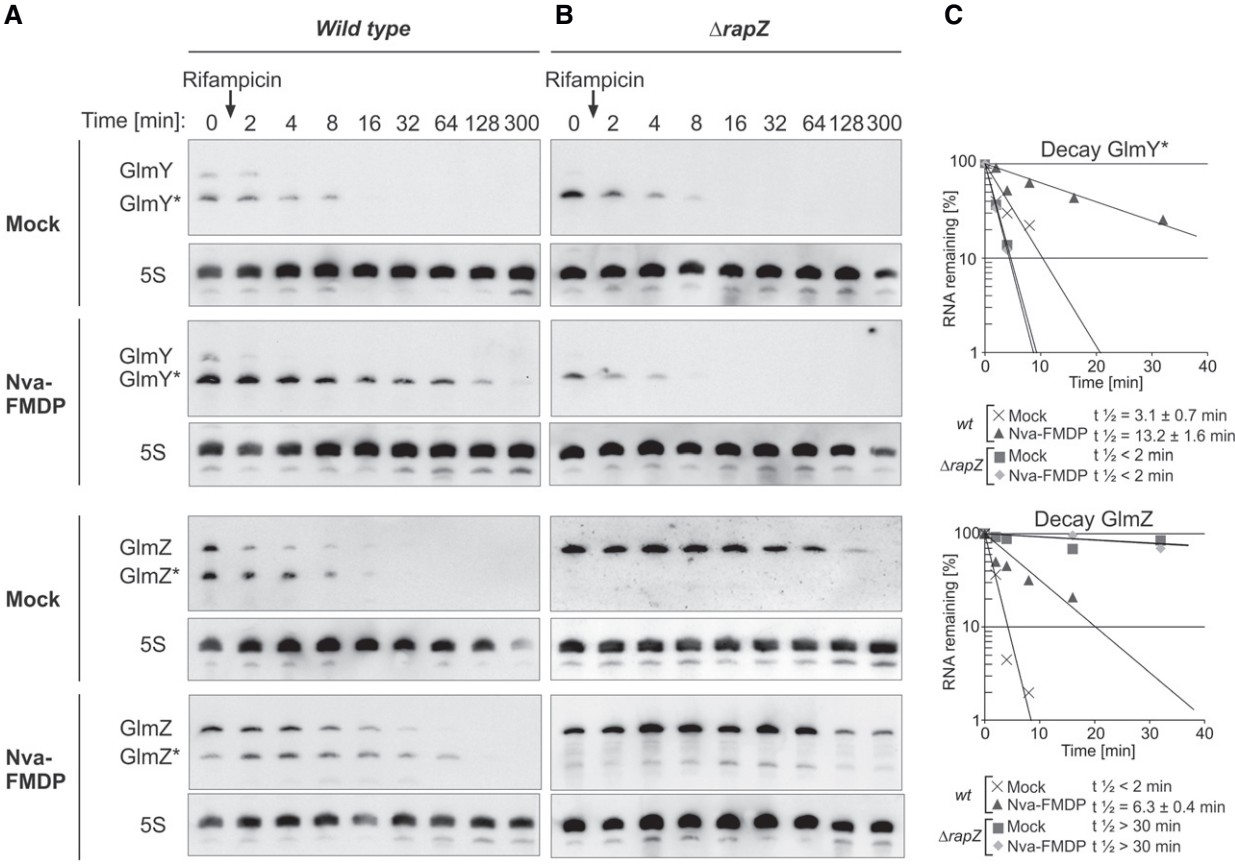

**Figure 5. GlmY* half-life increases under GlcN6P starvation conditions and RapZ is required for this effect.**

Northern blot experiments assessing the half-lives of GlmY* and GlmZ under normal growth and GlcN6P starvation conditions. Bacterial cultures were treated with either 100 μg/ml Nva-FMDP or $H_2O$ (mock). Transcription was stopped by rifampicin addition when cultures attained $OD_{600} = 1.0$ and samples were removed at indicated times for Northern analysis.

A  Analysis of GlmY* and GlmZ decay in the *wild-type* strain Z8.

B  Analysis of GlmY* and GlmZ decay in the *ΔrapZ* mutant Z28.

C  Semi-logarithmic plots of GlmY* and full-length GlmZ decay for half-life determination.

Data information: In (C), data are presented as mean, $n = 2$. Half-lives are presented as mean ± SD where applicable.

Source data are available online for this figure.

RapZ_quad retains the capability to activate *glmY* expression when tested in a complementation assay (Fig 2E), indicating that these functions can be separated. Plasmids expressing *rapZ_quad* or *wild-type rapZ* were introduced into a *ΔglmS* mutant strain lacking endogenous *rapZ*. Following induction of *rapZ* expression, cells were subjected to GlcN6P replete and depletion regimes and GlmY* half-life was determined. In the cells producing *wild-type* RapZ, half-life of GlmY increased ~2-fold upon GlcN6P depletion (Fig EV4A–C). Interestingly, in cells producing the RapZ_quad variant, GlmY* half-life remained short ($t_{1/2} \sim 3$ min) and unaffected by GlcN6P starvation as observed in the non-complemented *ΔrapZ* mutant (cf. Appendix Fig S12B). Thus, the RNA-binding function of RapZ is critical for protecting GlmY* from decay during GlcN6P starvation stress.

### GlcN6P controls RapZ/GlmY complex formation

Under GlcN6P limitation, GlmY* is stabilized and protected from degradation, likely by forming stable complexes with RapZ. This observation suggested that binding of GlmY* by RapZ is controlled by GlcN6P, which we investigated using EMSA.

As observed in previous studies (Göpel *et al*, 2013; Gonzalez *et al*, 2017), radiolabeled GlmY* was readily bound when incubated with increasing concentrations of RapZ (Fig 6D, left panel, lanes 1–4). Interestingly, GlmY* remained unbound when 7.5 mM GlcN6P was included in the assay (Fig 6D, left panel, lanes 5–7). Next, saturating concentrations, i.e., 1,200 nM of RapZ, were co-incubated with GlmY*, whereas the GlcN6P concentration was variable (Fig 6E). The presence of ≥ 5 mM GlcN6P prevented binding of GlmY* completely. At lower concentrations, a fraction of GlmY* remained unbound, whereas another fraction remained in the gel pocket. Albeit the nature of these latter complexes remains unclear, it appears that RapZ responds to GlcN6P concentrations as low as 1 mM, which are sufficient to prevent formation of canonical GlmY*/RapZ complexes. The ability to antagonize RapZ/GlmY* complex formation is a specific feature of GlcN6P as closely related metabolites including glucosamine-1-phosphate (GlcN1P), Glc6P,

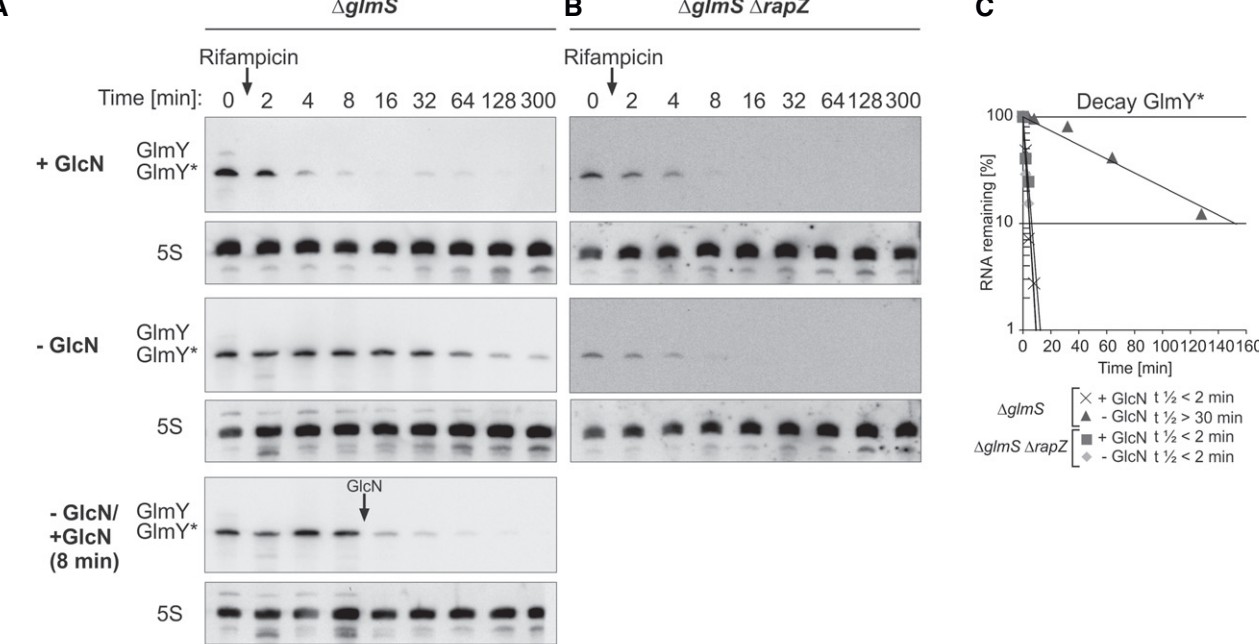

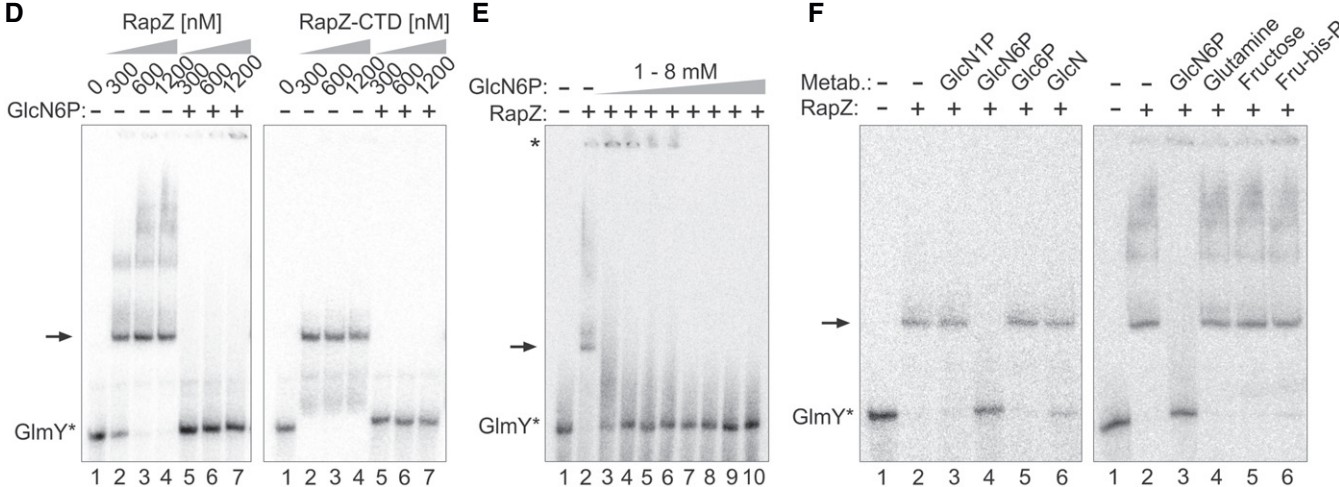

**Figure 6. GlcN6P releases GlmY* from RapZ leading to rapid degradation of the sRNA.**

Northern blot experiments (A–C) assessing GlmY* half-life under GlcN6P replete (+GlcN) and depletion conditions (−GlcN). Transcription was stopped by addition of rifampicin, and samples were harvested at indicated times for Northern analysis.

A  Analysis of the *ΔglmS* strain Z1126. The GlcN6P depleted culture was split 8 min after rifampicin addition and one of the sub-cultures was resupplied with GlcN (indicated by arrow).

B  The *ΔglmS ΔrapZ* double mutant Z1127 was tested.

C  Semi-logarithmic plots of GlmY* decay for half-life determination.

D  EMSA experiments addressing the role of GlcN6P for GlmY*/RapZ interaction. Radiolabeled GlmY* was incubated with incremental concentrations of RapZ (left panel) or RapZ-CTD (right panel) in the absence or presence of 7.5 mM GlcN6P. Binding reactions were separated on native PAA gels and analyzed by phospho-imaging. The RapZ/GlmY* complex is indicated by an arrow.

E  EMSA following incubation of GlmY* with 1,200 nM RapZ in the presence of various GlcN6P concentrations ranging from 0 (lane 2) to 8 mM (lane 10). The fraction of GlmY* remaining in the gel pocket is marked with an asterisk. The RapZ/GlmY* complex is indicated by an arrow.

F  EMSA following incubation of GlmY* with 1,200 nM RapZ in the absence or presence of 7.5 mM of the indicated metabolite. The RapZ/GlmY* complex is indicated by an arrow.

Data information: In (C), data are presented as mean, *n* = 2.
Source data are available online for this figure.

and GlcN had no effect (Fig 6F). Likewise, glutamine and fructose-1,6-bisphosphate, which represent global regulatory metabolites (Chubukov *et al*, 2014), had no role (Fig 6F).

### GlcN6P controls interaction of RapZ with GlmY by binding to the C-terminal domain

Previous work has shown that the RapZ-CTD dimer is capable of binding GlmY on its own (Gonzalez *et al*, 2017). Interestingly, the CTD contains a pocket composed of five residues located close to the RBD and potentially suited to accommodate a metabolite. Indeed, SPR spectroscopy revealed that the CTD binds GlcN6P on its own, whereas the NTD lacks this activity, supporting this idea (Fig 1C, bottom panels). Interestingly, the overall affinity of the RapZ-CTD for GlcN6P ($K_D = 10$ nM) is one order of magnitude higher as compared to the full-length protein due to a higher association rate ($k_a = 2.2 \times 10^5$ M$^{-1}$ s$^{-1}$; $k_d = 2.3 \times 10^{-3}$ s$^1$). Finally, when tested by EMSA, the RapZ-CTD performed equally well as the full-length protein in responding to GlcN6P (Fig 6D, right panel). Thus, the NTD of RapZ is not required for sensing this metabolite and responding by release of GlmY. We conclude that the RapZ-CTD carries out this function.

## Discussion

In this study, we identify the RBP RapZ being at the heart of bacterial cell envelope precursor metabolite sensing and signaling (Fig 7). Rather than to sense GlcN6P directly through interaction with the enzyme (Mouilleron *et al*, 2012), *E. coli* employs protein RapZ for this task (Fig 1). GlcN6P regulates two activities of RapZ: It abrogates binding of GlmY and also the ability of RapZ to activate the TCS QseE/QseF (Figs 3F and 6). Upon GlcN6P limitation, RapZ accumulates in a metabolite-free state and activates QseE/QseF by interaction, demonstrating that an RBP can directly transfer information to a transcription factor (Figs 1B, 2, and 3). RapZ stimulates QseE autophosphorylation, thus increasing levels of phosphorylated QseF (Fig 3D and E), which in turn activates *glmY* transcription (Figs 2 and 3C). Thereby generated GlmY* subsequently sequesters RapZ into long-lasting complexes as reflected by the drastically increased stability of GlmY* (Figs 5 and 6A–C, and EV4). Consequently, full-length GlmZ accumulates and activates GlmS synthesis (Figs 5 and EV3, Appendix Fig S11). Thus, RapZ upregulates its decoy GlmY* to prevent itself from binding sRNA GlmZ (Fig 7). Once replenished, GlcN6P releases RapZ from complexes with GlmY*, which is in turn rapidly degraded due to lack of protection

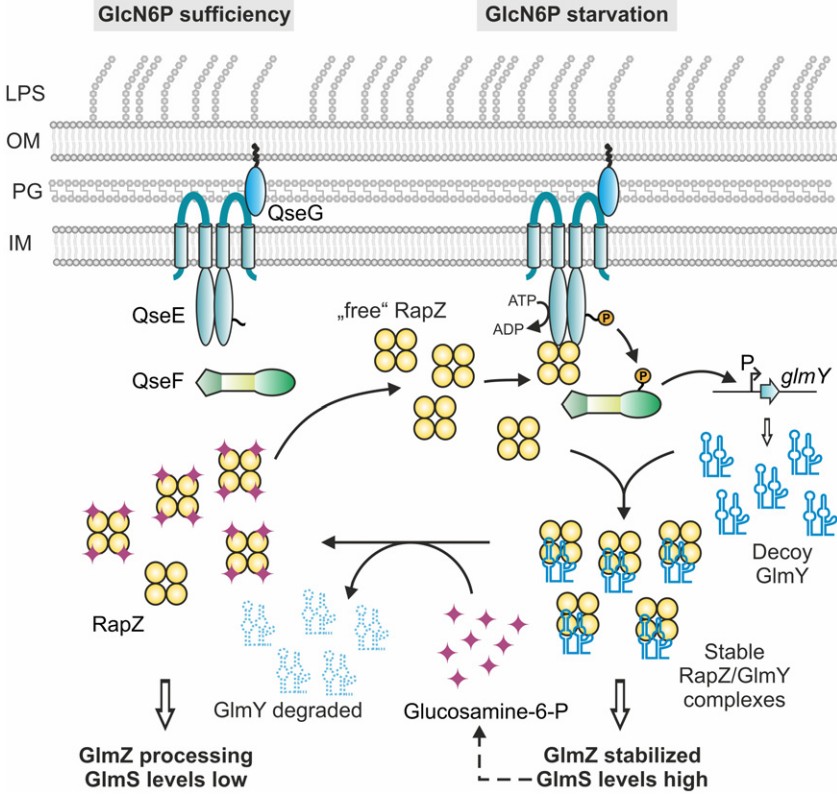

**Figure 7. Model for GlcN6P sensing and network control.**
Cartoon summarizing the current findings. RapZ, presumably in its tetrameric state, binds GlcN6P in its CTD. Upon GlcN6P scarcity, RapZ accumulates in its "free" form and activates phosphorylation of QseE/QseF by direct interaction. Activity of the TCS depends on lipoprotein QseG (Göpel & Görke, 2018), suggesting that RapZ can only activate those kinases, which are contacted by QseG in the periplasm. QseF~P triggers *glmY* expression from its σ⁵⁴ promoter augmenting levels of GlmY*, which subsequently sequesters RapZ into stable complexes. Consequently, RapZ is not available to trigger decay of sRNA GlmZ, which therefore activates synthesis of GlmS, replenishing GlcN6P. Sequestration also precludes RapZ from activating QseE/QseF, providing a negative feedback loop that adjusts GlmY amounts to the required level. GlcN6P releases GlmY* from RapZ, which is then free to promote GlmZ decay repressing *glmS*.

(Figs 5 and 6). In conclusion, the increase of GlmY* steady-state levels observed upon GlcN6P depletion in the current (Fig 1A) and previous work (Reichenbach *et al*, 2008, 2009; Khan *et al*, 2016) results from two distinct activities of RapZ: upregulation of *glmY* transcription through QseE/QseF and stabilization of GlmY* through its binding.

We further identify a negative feedback loop that limits *glmY* expression under GlcN6P starvation. Both GlmY and GlmZ bind RapZ and are thereby capable to counteract activation of QseE autophosphorylation (Fig 4C), limiting *glmY* expression (Fig 4A and B). Consequently, GlmY* levels will increase under GlcN6P starvation until all available RapZ is sequestered, preventing further stimulation of QseE/QseF. A similar feedback loop operates in the Csr circuitry, in which the RBP CsrA positively controls, albeit indirectly, synthesis of response regulator UvrY, which in turn activates expression of sRNA CsrB that counteracts CsrA activity by sequestration (Camacho *et al*, 2015). As a difference, GlmY directly interferes with communication between two signaling proteins to autoregulate its expression, which to the best of our knowledge represents a novel activity for an sRNA and also a novel mechanism for TCS feedback control (Groisman, 2016).

RapZ appears to activate QseE/QseF predominantly by stimulating QseE autophosphorylation (Fig 3). Additionally, RapZ also interacts with QseF, but whether it binds both proteins simultaneously or individually remains to be addressed. Interaction of RapZ with both QseE and QseF could help stabilize a ternary complex, compensating for the transient interactions with the individual proteins (Fig 3A and B). A scaffolding role for TCS signaling was also observed for the accessory protein UspC (Heermann *et al*, 2009). It will be interesting to learn whether the RBD of RapZ is also part of the surface contacting QseE/QseF, recapitulating the extensive overlap between RNA-binding and protein–protein interaction sites previously observed in human RBDs (Castello *et al*, 2016) also in bacteria. Nonetheless, both domains of RapZ and proper oligomerization are required for undisturbed binding of QseE/QseF suggesting that tetrameric RapZ activates this TCS (Fig EV1). Recently, we have shown that QseE/QseF employ the lipoprotein QseG as third essential component (Göpel & Görke, 2018). Kinase QseE requires interaction with QseG in the periplasm for activity suggesting that a quaternary signaling complex involving RapZ must form to obtain a fully activated TCS (Fig 7). Through integration of RapZ into this higher-order complex, QseE/QseF are recruited to GlcN6P starvation, providing the first identified *bona fide* stimulus for this TCS in *E. coli* K12. Perhaps, QseG monitors a process in the envelope to integrate information accordingly.

The RapZ-CTD binds GlmY and GlcN6P on its own (Figs 1C and 6D). In agreement, the CTD contains a pocket potentially suited to accommodate a metabolite, which is in close proximity to the RBD or could be even part of it (Gonzalez *et al*, 2017). As the RapZ-CTD responds on its own to GlcN6P by releasing GlmY (Fig 6D), GlcN6P and the sRNA may compete for access to the RapZ-CTD. The RapZ-NTD appears to have an auto-inhibitory role as reflected by the 10-fold higher affinity of the RapZ-CTD for GlcN6P as compared to the full-length protein (Fig 1C). Coinciding differences in affinities were also detected with respect to GlmY binding (Gonzalez *et al*, 2017), suggesting that the NTD limits access of both binding partners, perhaps in response to stimuli that remain to be identified. Interestingly, the putative GlcN6P binding site is highly conserved in RapZ homologs of diverse bacteria, whereas the RBD is restricted to *Enterobacteriaceae* coinciding with occurrence of GlmY/GlmZ (Göpel *et al*, 2013). It is tempting to speculate that control of TCS activity in response to metabolite binding is the evolutionary primordial function of RapZ, which could perhaps explain corresponding mutant phenotypes in species distantly related to *E. coli* (Luciano *et al*, 2009; Cui *et al*, 2018).

For *E. coli* cells grown in the absence of amino sugars, intracellular GlcN6P concentrations in the range from 62 μM to 1.15 mM were reported and may increase up to ~10 mM when cells grow on amino sugars (Bennett *et al*, 2009; Alvarez-Anorve *et al*, 2016). This concentration range fits well with our results indicating that 1–2 mM GlcN6P is necessary to elicit a response of RapZ *in vitro* (Figs 3F and 6E). Moreover, our *in vivo* data show that RapZ already stimulates QseE/QseF activity under normal growth conditions (Figs 2E and 3C), suggesting that a fraction of RapZ is not in complex with GlcN6P, but free to interact with QseE/QseF or the sRNAs. So far we have no explanation for the high affinity of RapZ for GlcN6P as measured by SPR ($K_D$ = 186 nM; Fig 1C). However, it is possible that the affinities of the individual pockets within the RapZ tetramer may change during their sequential occupation by GlcN6P. This could also include formation of mixed RapZ oligomers simultaneously binding GlcN6P and GlmY. Finally, our work leaves open, how GlcN6P interferes with binding of GlmZ by RapZ. Intuitively, GlcN6P bound RapZ should perform this task. On the other hand, a fraction of RapZ is apparently not in complex with GlcN6P under normal growth conditions and is therefore theoretically free to bind GlmZ.

RNA-binding domains whose RNA-binding activity is controlled through binding of a metabolite are not unprecedented in bacteria. Well-studied examples include antitermination proteins like HutP and TRAP that control expression of amino acid biosynthesis genes in response to availability of the cognate amino acid (Babitzke *et al*, 2019). Aconitase, an iron-containing enzyme of the Krebs cycle, is known to bind and regulate its own mRNA when iron is low and the enzyme accumulates in the apo-form (Benjamin & Masse, 2014). However, RapZ represents the first sRNA-binding protein responding directly to a metabolite. Furthermore, both functions of RapZ, sRNA GlmY binding and activation of QseE/QseF, are concurrently controlled by the metabolite.

Why does *E. coli* employ this complex circuitry to regulate a single mRNA? The factors involved might integrate additional signals (e.g., through QseG) and connect the GlcN6P signal with further targets such as virulence genes in pathogens (Lustri *et al*, 2017). Moreover, RapZ could recruit the $\sigma^E$ response to GlcN6P starvation, as QseF may also activate a promoter upstream of *rpoE* (Klein *et al*, 2016). This could help to overcome envelope stress caused by precursor depletion. Recently, GlmS in *Salmonella* was reported to be inhibited by protein PtsN in response to glutamine and GlcN6P levels (Yoo *et al*, 2016). How GlcN6P is sensed by PtsN is unknown, but our work uncovers RapZ as feasible candidate.

## Materials and Methods

### Strains, plasmids, oligonucleotides, and growth conditions

Strains and plasmids are listed in Appendix Tables S1 and S2, and their construction is described in the Appendix Supplementary

Methods section. Oligonucleotides are documented in Appendix Table S3. Bacteria were routinely grown in LB medium at 37°C. If required, antibiotics were added at following concentrations: ampicillin (100 μg/ml), chloramphenicol (15 μg/ml), kanamycin (30 μg/ml), spectinomycin (50 μg/ml), and tetracycline (12.5 μg/ml). Growth of *ΔglmS* strains was sustained by supplementing LB medium with 0.2% GlcN. For experiments assaying GlcN6P replete and depletion conditions, an overnight culture of the *ΔglmS* mutant was inoculated to an $OD_{600} = 0.1$ and grown until $OD_{600} = 0.3$. Subsequently, cells were washed and split into two cultures, one of which lacked GlcN. Growth was continued, and samples were harvested hourly for analysis of RNA steady-state levels or until $OD_{600} = 1.0$ for determination of sRNA half-life. Similarly, when using Nva-FMDP, a pre-culture was split into two cultures at $OD_{600} = 0.3$ and one of the cultures was provided with the required Nva-FMDP concentration (1 mg/ml stock solution), whereas $H_2O$ was added to the other culture. Subsequently, growth was continued and cells were harvested as described for the *ΔglmS* mutant.

### RNA extraction, Northern Blotting, and sRNA half-life determination

Total RNA was extracted and analyzed by Northern blotting as described recently (Durica-Mitic & Görke, 2019). For RNA half-life determinations, 500 μg/ml rifampicin was added to the cultures when reaching $OD_{600} = 1.0$ and aliquots were harvested at indicated times by pelleting and freezing in liquid nitrogen. RNA signal intensities were normalized to 5S signals and plotted semi-logarithmically in percent against time. The resulting graphs present the average values of at least two independent experiments.

### Determination of β-galactosidase activity

β-Galactosidase activities of cells were determined as previously described (Miller, 1972). To economize usage of Nva-FMDP, activities were also determined from cultures grown in 96-well plates. To this end, a 10 ml culture was grown in a flask until $OD_{600} = 0.3$. Subsequently, 270-μl aliquots were distributed to wells of a 96-well plate containing the required Nva-FMDP amounts in 30 μl $H_2O$ to obtain a final culture volume of 300 μl per well. Growth of cultures and $OD_{600}$ recordings were performed using the Synergy H1 microplate reader (BioTek). 50-μl aliquots were harvested at hourly intervals for determination of β-galactosidase activity.

### Protein purification

Protein purification procedures are described in the Appendix Supplementary Methods section.

### Detection of GlcN6P by HILIC-MS/MS (targeted metabolomics)

Metabolites were extracted from the protein elution fractions, and a targeted metabolomics approach using HILIC-MS/MS (Virgiliou *et al*, 2018) was employed for identification of GlcN6P. For metabolite extraction, the Strep-Tactin affinity chromatography-derived eluates containing ~100 μg protein were mixed with 200 μl methanol and incubated at −20°C for 2 h, respectively. After

centrifugation and transferring the supernatant to a new tube, the pellet was incubated first with 200 μl methanol and subsequently with 200 μl acetonitrile. Following centrifugation, both supernatants were combined, and the solvent was evaporated in a vacuum centrifuge. After resolving the extracted metabolites in 50 μl of a 1:1 mixture of 10 mM ammonium acetate and acetonitrile, 10 μl were subjected to LC-MS/MS, which was performed using a TSQ Quantiva triple quadrupole mass spectrometer (Thermo Scientific) coupled to an UltiMate 3000 RSLC system (Dionex, Thermo Scientific). Metabolites were separated using a gradient between 5% mobile phase A (10 mM ammonium acetate in water; pH 7.0) to 50% A in phase B (acetonitrile) using a ZIC-HILIC column (SeQuant® ZIC®-HILIC 3.5 μm, 200 Å, 100 × 2.1 mm) employing a flow rate of 100 μl/min. For identification of the metabolite, multiple transitions were used, each of which is a set of three important parameters: (i) the mass/charge ratio of the metabolite, (ii) a specific fragmentation potential, and (iii) the mass/charge ratio of one fragment of the metabolite. Metabolites were quantified using selected reaction monitoring (SRM) in the negative ion mode, employing the transitions 258.1 m/z to 97 m/z for quantitation and 258.1 m/z to 79 m/z as qualifier. Each experiment was performed in duplicate.

### Surface plasmon resonance (SPR) spectroscopy

Surface plasmon resonance assays were performed in a Biacore T200 (GE Healthcare, Freiburg) using carboxymethyl dextran sensor chips (Sensor Chip Series S CM5, GE Healthcare) that were previously coated with Strep-Tactin® XT resin (IBA, Göttingen). For that purpose, the chips were equilibrated with HBS-EP buffer [10 mM HEPES (pH 7.4), 150 mM NaCl, 3 mM EDTA, 0.005 % (v/v) detergent P20] until the dextran matrix was swollen. Subsequently, the flow cells were activated by injecting a 1:1 mixture of N-ethyl-N-(3-dimethylaminopropyl) carbodiimide hydrochloride and N-hydroxysuccinimide using the standard amine-coupling protocol at a flow rate of 10 μl/min. All flow cells were then loaded with a final concentration of 10 μg/ml of Strep-Tactin® XT resin in 10 mM acetate pH 5.5 using a contact time of 420 s, so that the surfaces contained densities of 5,000–6,000 resonance units (RU). Free binding sites of the flow cells were saturated by injection of 1 M ethanolamine/HCl pH 8.0. Interaction of Strep-RapZ, Strep-RapZ-NTD, or Strep-RapZ-CTD with metabolites was analyzed using a single-cycle kinetics approach with running buffer [10 mM Tris/HCl (pH 7.0), 100 mM NaCl, 10 mM $MgCl_2$ 0.005% (w/v) P20]. Strep-tagged proteins (20 μg/ml) were captured onto the chip using a contact time of 180 s at a constant flow rate of 10 μl/min followed by a stabilization time of 180 s, resulting in capturing of ~2,500 RU of the respective Strep-RapZ variant. Single-cycle kinetics using the metabolites were performed at a flow rate of 30 μl/min. Increasing concentrations (100, 500, 1,000, 2,500, 5,000 nM) of the respective metabolite were sequentially injected onto the flow cells without interim regeneration using a contact time of 180 s each and a final dissociation time of 180 s. Then, the chip was regenerated by injection of 10 mM glycine pH 1.5 for 60 s at a flow rate of 30 μl/min over all flow cells, which completely removed the Strep-tagged protein from the surface. Furthermore, blank single-cycle kinetics were recorded by sequentially injecting running buffer instead of increasing metabolite concentrations.

Binding between RapZ and QseE and QseF was assessed in a multicycle approach in HBS-EP buffer using carboxymethyl dextran sensor chips previously coated with Strep-Tactin® XT resin (see above). Strep-RapZ (10 µg/ml) was captured onto the chip using a contact time of 60 s at a constant flow rate of 10 µl/min followed by a stabilization time of 300 s. Thereby, 200–400 RU of Strep-RapZ was captured onto the chip. Then, increasing concentrations (10 nM, 50 nM, 2 × 100 nM, 250 nM, 500 nM, and 1,000 nM) of QseE-His$_{10}$ or QseF-His$_{10}$ were injected over the chip surface for 180 s at a flow rate of 30 µl/min, followed by a dissociation time of 360 s. After each cycle, the chip was regenerated by removing Strep-RapZ from the surface by injection of 10 mM glycine pH 1.5 for 60 s at a flow rate of 30 µl/min.

All experiments were performed at 25°C. Sensorgrams were recorded using the Biacore T200 Control software 2.0 and analyzed with the Biacore T200 Evaluation software 2.0 or TraceDrawer software 1.8.1 (Ridgeview Instruments AB, Uppsala, Sweden). The surface of flow cell 1 was used to obtain blank sensorgrams for subtraction of bulk refractive index background. Buffer controls on the second surface were subtracted from the sensorgrams obtained with Glc6NP, GlcN, or Glc6P, respectively, to normalize drifts on the surface. The referenced sensorgrams were then normalized to a baseline of 0. Peaks in the sensorgrams at the beginning and the end of the injections emerged from the runtime difference between the flow cells of each chip. Shown sensorgrams represent one characteristic of three independently performed experiments.

### Bacterial adenylate cyclase-based two-hybrid (BACTH) assay

The BACTH assay is based on the reconstitution of adenylate cyclase activity in *E. coli* strains lacking the endogenous gene (Karimova *et al*, 1998). Reconstitution occurs through interaction of proteins fused to the complementary T18 and T25 domains of the *Bordatella pertussis* adenylate cyclase, leading to cAMP synthesis. Protein interaction is quantified by measuring β-galactosidase activity, whose synthesis depends on intracellular cAMP-CRP levels. The reporter strain (RH785 or BTH101) carrying derivatives of plasmids pUT18C and pKT25 encoding the desired T18 and T25 fusion constructs was grown at 28°C in the presence of 1 mM IPTG, and the β-galactosidase activities were determined from cells grown to stationary phase.

### QseE/QseF *in vitro* phosphorylation assay

Kinase assays were performed in total volumes of 10–30 µl, depending on the number of aliquots to be analyzed over time. 1 µM QseE'-His$_{10}$ was pre-incubated in reaction buffer (50 mM Tris–HCl, pH 7.6, 200 mM KCl, 10 mM MgCl$_2$, 5 mM MnCl$_2$) for 5 min at 25°C in the absence or presence of 5 µM Strep-RapZ. For analysis of QseE'-His$_{10}$ autophosphorylation, 10 µCi [$\gamma$-$^{32}$P]-ATP (Hartmann Analytic) and 100 µM cold ATP were added and aliquots were removed at indicated times. To analyze phosphoryl-group transfer from QseE'-His$_{10}$ to QseF-His$_{10}$, QseE'-His$_{10}$ was incubated with [$\gamma$-$^{32}$P]-ATP/ATP for 10 min and subsequently 1 µM QseF-His$_{10}$ was added. Aliquots were removed at indicated times and mixed with 2 × SDS sample buffer to stop reactions. The proteins were separated by SDS–PAGE and subsequently analyzed by using a Typhoon FLA-9500 phospho-imager (GE Healthcare).

### *In vitro* transcription and radioactive labeling of sRNA

A description of *in vitro* transcription and labeling of sRNAs is provided in the Appendix Supplementary Methods section.

### EMSA

Binding reactions were performed in 1 × binding buffer (10 mM Tris–HCl, 100 mM KCl, 10 mM MgCl$_2$) in a volume of 10 µl. Strep-RapZ or Strep-RapZ-CTD was serially diluted in 5 µl 1 × binding buffer and incubated with the respective metabolite for 15 min at 30°C. Radiolabeled GlmY* was mixed in 5 µl 1 × binding buffer with 1 µg of yeast tRNA (Ambion), heat-denatured, chilled, and subsequently added to the protein/metabolite mixtures. Following an additional incubation for 30 min at 30°C, 5 × native loading buffer (50% glycerol, 0.5 × TBE, 0.2% bromophenol blue) was added and samples were separated on native gels (5.5% PAA, 1 × TBE) at 4°C using 0.5 × TBE as running buffer. Gels were analyzed by phospho-imaging (Typhoon FLA 9000, GE Healthcare). It should be noted that the outcome of the EMSA assays is pH-dependent, as GlcN6P had no effect when reactions were performed at pH $\geq$ 8.0 (Appendix Fig S14). This can at least partially be explained by the interconversion of GlcN6P between two anomeric forms, the ratio of which is pH-dependent. Generally, the α anomer, whose formation is favored at pH < 7.0, appears to be physiologically relevant, e.g., the β-anomer lacks the ability to activate the *glmS* ribozyme (Davis *et al*, 2011). To ensure that the α-anomer prevails, the binding reactions analyzed in Fig 6D–F were conducted at pH 6.0. At pH 7.0, a fraction of GlmY* was not released but remained in the gel pocket (Appendix Figs S14 and S15). Whether the latter is a consequence of the additional presence of the β-anomer, results from the theoretic ability of the RapZ tetramer to form continuous polymers (Gonzalez *et al*, 2017) and/or represents RapZ oligomers binding GlcN6P and GlmY simultaneously, remains to be clarified.

### Statistics

β-Galactosidase assays were performed using at least three biological replicates except for Fig 1A and Appendix Figs S2 and S6 (*n* = 2). All other experiments were carried out at least two times, and representative results are shown. Key experiments including Fig 1A, BACTH data (Fig 3A), and sRNA half-life determinations were reproduced using independent *E. coli* strain lineages (*n* = 2; data in Appendix). Two-tailed Student's *t*-test was performed to assess whether two data sets are significantly different. The calculated *P* values are reported in the source data files.

Expanded View for this article is available online.

### Acknowledgements

We thank Lena Hoffmann, Anna Kögler, Denise Lüttmann, and Birte Reichenbach for help with construction of strains or plasmids. We thank Régis Hallez for strain RH785 and Ryszard Andruszkiewicz and Slawomir Milewski for generously providing Nva-FMDP. HILIC-MS/MS was performed at the VBCF Metabolomics unit, which is supported by the Vienna Business Agency. We thank Kirsten Jung for access to the Bioanalytics core facility of the Biocenter at Ludwig-Maximilians-University Munich to perform the SPR analyses. We

thank Ben Luisi and Jörg Vogel for comments on the manuscript. This work was supported by stand-alone grants P26681-B22 and P32410-B of the "Austrian Science Fund" (FWF) to B.G. and the Doktoratskolleg RNA Biology W1207-B09.

## Author contributions

BG conceived and designed the study. MAK, SD-M, and YG designed and performed experiments. RH performed SPR spectroscopy. All authors analyzed data. MAK and BG wrote the paper with contributions of YG and RH.

## Conflict of interest

The authors declare that they have no conflict of interest.

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
