## [Review Process File · The EMBO Journal]

Small RNA-binding protein RapZ mediates cell envelope precursor sensing and signaling in *Escherichia coli*

Muna A. Khan, Svetlana Durica-Mitic, Yvonne Göpel, Ralf Heermann and Boris Görke

Review timeline:

Submission date:	28th Oct 2019
Editorial Decision:	18th Nov 2019
Revision received:	9th Jan 2020
Editorial Decision:	20th Jan 2020
Revision received:	21st Jan 2020
Accepted:	24th Jan 2020

Editor: Ieva Gailite

Transaction Report:

1st Editorial Decision

18th Nov 2019

Thank you for submitting your manuscript for consideration by The EMBO Journal. We have now received three referee reports on your manuscript, which are included below for your information.

As you will see from the comments, all reviewers appreciate the work and the quality of the data and recommend publication of the manuscript after a minor revision. Given these positive evaluations from three experts of the field, I would like to invite you to submit a revised version of your manuscript in which you address the issues raised in reviewers' reports.

REFeree REPORTS:

Referee #1:

In this study, the authors report major advances in understanding the regulatory circuitry by which *E. coli* maintains an appropriate rate of synthesis for GlcN6P, key precursor of cell envelope polymers. This work builds on an elegant series of studies by this group, which defined the posttranscriptional regulation of *glmS* (glucosamine-6-P synthase) expression via an RNA binding protein, RapZ, which uses two noncoding sRNAs to mediate and control its activity. The sRNA GlmZ pairs with and facilitates translation of *glmS*. This sRNA is subject to turnover directed by binding to RapZ. GlmY sRNA acts as a decoy that also binds to RapZ but is not subsequently degraded, thus sparing GlmZ under conditions of low GlcN6P. The cause of GlmY accumulation in response to GlcN6P levels has remained a central unexplained element of the model. Here, the authors demonstrate that RapZ binding to GlcN6P itself controls its ability to bind to GlmY and to activate a two-component signal transduction system (QseE/F), leading to transcription of GlmY. Only the GlcN6P-free form RapZ was capable of binding to and activating the kinase activity of QseE. GlmY binding to RapZ was also shown to prevent RapZ from activating QseE/F, thus generating a feedback loop that limits the response. These findings and a number of others, examined in rigorous detail, make for fascinating biological story that highlights a number of novel

mechanistic features for a bacterial RNA binding protein. This study represents a tour-de-force for the Görke group, and the findings will be of interest to a broad cadre of scientists studying basic RNA-protein mechanisms, regulation of gene expression, bacterial physiology, cell envelope homeostasis, etc.

The writing is clear, concise and truly a pleasure to read. The claims are supported by the evidence presented and the treatment of historical context is appropriate. The supplementary data are valuable to support the findings.

My only technical suggestion is to describe the statistical analyses, including the meaning of the error bars shown in several of the figures.

As a minor point for the authors' discretion: is there any relevance for the occurrence of biphasic RNA decay patterns that are seen in some cases, e.g. Fig. S13, S16?

Referee #2:

This manuscript unexpectedly reveals that the key sRNA-binding protein RapZ is also capable to respond to the intracellular concentration of the metabolite glucosamine 6-phosphate (GlcN6P). By using various mutant strains and *in vivo* and *in vitro* approaches, the authors have demonstrated the following data: (1) RapZ specifically binds to GlcN6P using its CTD domain. (2) At low levels of GlcN6P, the free RapZ stimulates the phosphorylation of the two-component system QseE/QseF through specific but transitory binding, which contribute to enhance GlmY levels. (3) In turn, GlmY sequesters RapZ to prevent GlmZ decay and concomitantly favors GlmS synthesis. (4) The level of GlmY controls the availability of RapZ introducing a feedback regulatory loop. (5) When the concentration of GlcN6P enhances, the metabolite competes with GlmY for binding to RapZ. In turn, GlmY is rapidly degraded and GlmZ is further processed and cannot be used anymore as an activator of glmS translation. This is a remarkable study, which illustrates the diversity of functions of a key RNA-binding protein and the complexed regulatory circuit regulating glmS expression.

Only minor comments are addressed as follows.

1- Figure 6D: Addition of increasing concentrations of RapZ to labeled GlmY showed the formation of several complexes. Because RapZ is a tetramer, it is expected that several molecules of GlmY would bind to RapZ. Using SPR methodology, I assume that the authors should be able to monitor the stoichiometry of the GlmY-RapZ complex. Depending on the intracellular concentrations of RapZ and GlmY, it is expected that various types of complexes might form *in vivo*.

2- Figures S17 and 6E: It is surprising that at a concentration of GlcN6P below 2 mM, GlmY formed high molecular weight complex with RapZ. Do the authors totally exclude that RapZ might simultaneously bind to GlcN6P and GlmY?

3- The authors use *in vitro* a recombinant RapZ protein for *in vitro* assays. At which extremity of the protein the tag was added? Did the authors check that the tag (Strep or His) did not alter the properties of RapZ and the formation of its tetrameric structure. Is Strep-RapZ able to complement a Δ rapZ mutant strain?

4- The biacore analysis showed that RapZ binds weakly to the isolated QseE/F proteins with high dissociation rates. However, it is not clear whether RapZ bind simultaneously to QseE and QseF.

Referee #3:

In this study, Khan et al. set out to determine how cells are measuring intracellular glucosamine-6-phosphate (GlcN6P) to control the levels of the GlmY sRNA. The authors report that the RapZ RNA-binding protein itself binds GlcN6P and that the binding of GlcN6P controls the ability of RapZ to bind and activate the activity of the QseE-QseF two component system. Activated QseF response regulator activates glmY transcription. This forms a nice auto regulatory loop in which more GlmY titrates RapZ away from the GlmZ small RNA, leading to more GlmZ and then more GlmS protein and finally more GlcN6P, which in turn binds RapZ to prevent it from activating QseE-QseF as well as from binding GlmY.

The study includes a number of different approaches to test the proposed model, and the manuscript is carefully prepared.

I only have one major comment:

1. Since the concept that the RNA-binding protein RapZ also binds the GlcN6P small molecule as well as both the QseE sensor kinase and QseF response regulator is quite novel, I would love to see a little more analysis (mutational or crosslinking) to delineate the regions of the RapZ C-terminal domain that contact all of these different molecules.

More minor comments:

2. Figure 6E: This EMSA image is not ideal.

3. Since the regulatory loop and the details of some of the experiments are challenging to keep track of, the writing needs to be as clear as possible. Two sentences that were confusing to me:
 -Page 5, line 5 from bottom: "Importantly, Nva-FMDP had no role for GlmY* in the Δ rapZ mutant."
 -Page 11, line 1 from top: "To survey these results,..."
 -Page 14, line 5 from top: "In this study, we identify the RBP RapZ..." (this sentence seems incomplete).

4. Suggestions:

-Page 10, line 7 from bottom: Suggest "is capable of limiting its own".
 -Figure 1C: check heading of bottom panel.
 -Figures 3, 4 and 6: Suggest adding lane numbers to figures, since the authors refer to specific lanes in the legend.

1st Revision - authors' response

9th Jan 2020

Revision of manuscript EMBOJ-2019-103848 entitled "*A small RNA-binding protein dedicated to cell envelope precursor sensing and signaling in Escherichia coli*".

Point-by-point response to reviewers' comments

We thank all referees for their encouraging comments and their thorough evaluation of our work. We are very grateful for their constructive criticisms, which we took very seriously. We also performed some additional experiments, which have now been incorporated into the manuscript and/or are discussed in the point-by-point response below. The novel results further support our conclusions and we believe that the manuscript is now significantly improved.

Referee #1:

In this study, the authors report major advances in understanding the regulatory circuitry by which *E. coli* maintains an appropriate rate of synthesis for GlcN6P, key precursor of cell envelope polymers. This work builds on an elegant series of studies by this group, which defined the posttranscriptional regulation of glmS (glucosamine-6-P synthase) expression via an RNA binding protein, RapZ, which uses two noncoding sRNAs to mediate and control its activity. The sRNA GlmZ pairs with and facilitates translation of glmS. This sRNA is subject to turnover directed by binding to RapZ. GlmY sRNA acts as a decoy that also binds to RapZ but is not subsequently degraded, thus sparing GlmZ under conditions of low GlcN6P. The cause of GlmY accumulation in response to GlcN6P levels has remained a central unexplained element of the model. Here, the authors demonstrate that RapZ binding to GlcN6P itself controls its ability to bind to GlmY and to activate a two-component signal transduction system (QseE/F), leading to transcription of GlmY. Only the GlcN6P-free form RapZ was capable of binding to

and activating the kinase activity of QseE. GlmY binding to RapZ was also shown to prevent RapZ from activating QseE/F, thus generating a feedback loop that limits the response. These findings and a number of others, examined in rigorous detail, make for fascinating biological story that highlights a number of novel mechanistic features for a bacterial RNA binding protein. This study represents a tour-de-force for the Görke group, and the findings will be of interest to a broad cadre of scientists studying basic RNA-protein mechanisms, regulation of gene expression, bacterial physiology, cell envelope homeostasis, etc.

The writing is clear, concise and truly a pleasure to read. The claims are supported by the evidence presented and the treatment of historical context is appropriate. The supplementary data are valuable to support the findings.

My only technical suggestion is to describe the statistical analyses, including the meaning of the error bars shown in several of the figures.

The data are presented as mean \pm SD. We added this information as well as the number of independent biological experiments underlying the mean values to the corresponding figure legends. We performed two-tailed student's t-test to assess whether two data sets are significantly different. The calculated p values are reported in the source data files. See also "author checklist" and the added "Statistics" statement in Materials and Methods section.

As a minor point for the authors' discretion: is there any relevance for the occurrence of biphasic RNA decay patterns that are seen in some cases, e.g. Fig. S13, S16?

To account for the pronounced stabilization of the sRNAs under GlcN6P depletion conditions, we monitored their fates over a long period of time. Hence, it is possible that sRNA degradation decelerated towards the end of the observation period as dedicated RNases may have turned over but could not be re-synthesized. This could explain why in some decay experiments the last data point was located above the trend line as noted by this reviewer. To obtain more accurate results, we omitted the last time point when located above the trend line and re-calculated corresponding decay plots and half-lives. We applied this re-analysis to Fig 5C, Appendix Fig S10C (formerly Supplemental Fig. S11C), Appendix Fig S11 (formerly Supplemental Fig. S13) and Fig EV4 (formerly Supplemental Fig. S16). In any case, our conclusions are not affected by the slightly changed decay plots and half-lives. We thank the reviewer very much for pointing out this observation.

Referee #2:

This manuscript unexpectedly reveals that the key sRNA-binding protein RapZ is also capable to respond to the intracellular concentration of the metabolite glucosamine 6-phosphate (GlcN6P). By using various mutant strains and in vivo and in vitro approaches, the authors have demonstrated the following data: (1) RapZ specifically binds to GlcN6P using its CTD domain. (2) At low levels of GlcN6P, the free RapZ stimulates the phosphorylation of the two-component system QseE/QseF through specific but transitory binding, which contribute to enhance GlmY levels. (3) In turn, GlmY sequesters RapZ to prevent GlmZ decay and concomitantly favors GlmS synthesis. (4) The level of GlmY controls the availability of RapZ introducing a feedback regulatory loop. (5) When the concentration of GlcN6P enhances, the metabolite competes with GlmY for binding to RapZ. In turn, GlmY is rapidly degraded and GlmZ is further processed and cannot be used anymore as an activator of glmS translation. This is a remarkable study, which illustrates the diversity of functions of a key RNA-binding protein and the complexed regulatory circuit regulating glmS expression.

Only minor comments are addressed as follows.

1- Figure 6D: Addition of increasing concentrations of RapZ to labeled GlmY showed the formation of several complexes. Because RapZ is a tetramer, it is expected that several molecules of GlmY would bind to RapZ. Using SPR methodology, I assume that the authors should be able to monitor the stoichiometry of the GlmY-RapZ complex. Depending on the intracellular concentrations of RapZ and GlmY, it is expected that various types of complexes might form *in vivo*.

The RapZ-CTD, which dimerizes and binds RNA on its own, forms a single complex with GlmY (cf. Fig. 6D, right panel and Appendix Fig S15A, right panel) indicating that it binds one molecule of the sRNA. In this respect, it is reasonable to assume that the RapZ tetramer binds up to two GlmY molecules, which might explain the additional super-shift observable in our EMSA experiments. Unfortunately, SPR spectroscopy is not suited to confirm this reasoning unambiguously. Capturing of Strep-RapZ onto the sensor chip could result in steric hindrance, selectively blocking interaction of some protomers within the RapZ tetramer with the sRNA. Such a steric inhibition, which has previously been observed in SPR analyses of heteromeric protein complexes (cf. Berkowitz et al., 2002, *J. Biol. Chem.* 277, 30629-30634), would consequently lead to an overestimation of the RapZ:GlmY ratio in the complex (e.g. 4:1 rather than 4:2).

In any case, we feel that clarification of the stoichiometry of the RapZ/GlmY complex is not directly relevant for our study. At concentrations ≥ 2 mM, GlcN6P hinders formation of all RapZ/GlmY complexes irrespective of their individual stoichiometry (Fig 6D and Fig 6E). At intermediate GlcN6P concentrations (1 mM), formation of mixed RapZ oligomers binding GlmY and GlcN6P simultaneously might be possible (see minor comment 2 below), which is now mentioned in the discussion and the Materials and Methods section (p. 16 and p. 20).

2- Figures S17 and 6E: It is surprising that at a concentration of GlcN6P below 2 mM, GlmY formed high molecular weight complex with RapZ. Do the authors totally exclude that RapZ might simultaneously bind to GlcN6P and GlmY?

We thank the reviewer for pointing out this interesting possibility. According to our data, GlcN6P and GlmY compete for getting access to the RapZ-CTD. Nonetheless, it cannot be excluded that at intermediate GlcN6P concentrations, RapZ may form mixed oligomers being simultaneously in complex with GlcN6P and GlmY. We now mention this possibility in the Discussion and the Materials and Methods sections (p. 16 and p. 20).

3- The authors use *in vitro* a recombinant RapZ protein for *in vitro* assays. At which extremity of the protein the tag was added? Did the authors check that the tag (Strep or His) did not alter the properties of RapZ and the formation of its tetrameric structure. Is Strep-RapZ able to complement a $\Delta rapZ$ mutant strain?

The Strep-tag is fused to the N-terminus of RapZ (now mentioned in the manuscript on p. 8) and does not interfere with known activities of RapZ. We have used Strep-RapZ already in previous work and demonstrated that it is able to complement a $\Delta rapZ$ mutant in respect to GlmZ processing and regulation of GlmS synthesis (Durica-Mitic and Görke, 2019, Fig. 1), binds GlmY and GlmZ *in vivo* by means of pull-down experiments (Göpel et al., 2013, Fig. 1A and B; Gonzalez et al., 2017, Fig. 7A), interacts with RNase E as observed in pull-down experiments (Göpel et al., 2013, Fig. 5B) and responds to GlcN6P depletion *in vivo* by switching sRNA binding partners (Göpel et al., 2013; Fig. 6E). As tetramerization is strictly required for RapZ activity *in vivo* (Gonzalez et al., 2017), one may conclude that Strep-RapZ is able to form a functional tetramer.

To rule out any remaining ambiguity, we added an experiment to Fig 2E demonstrating that Strep-RapZ can also activate *glmY* expression (Fig 2E last two columns, experiment mentioned on p. 8). For consistency, we measured all strains analyzed in Fig 2E in parallel once again.

4- The biacore analysis showed that RapZ binds weakly to the isolated QseE/F proteins with high dissociation rates. However, it is not clear whether RapZ bind simultaneously to QseE and QseF.

This is a challenging question difficult to tackle. As kinase QseE and response regulator QseF also interact with each other, classical protein-protein interaction assays such as pulldown or co-IP will always detect all three proteins (RapZ, QseE, QseF) in the output fractions regardless what the bait was. To obtain initial insight, we performed surface plasmon resonance experiments, in which we injected QseE and QseF proteins individually as well as in a 1:1 mixture onto the sensor chip coated with Strep-RapZ as ligand. The graph below depicts the obtained responses (RU units) as a function of the injected QseE and/or QseF concentrations. Interestingly, when injecting both proteins together at a concentration of 250 nM each, a much stronger response as observed with the individual proteins is obtained. The response generated by the QseF/QseE mixture (789,3 RU) is 3-fold higher than the sum of the responses obtained with the individual proteins (115,7 for QseF; 163,9 for QseE). This difference suggests a higher affinity of RapZ for the QseE/QseF complex as compared to the individual proteins, which supports the idea of a simultaneous interaction of RapZ with QseE and QseF. However, when injecting both QseE and QseF at higher concentrations, lower responses as observed for the individual proteins are recorded. One possible explanation is that QseF and/or QseE change their oligomeric state at higher concentrations, impairing their interaction and consequently also binding to captured Strep-RapZ. For instance, response regulators usually contact their cognate kinases in monomeric form, whereas the activated (phosphorylated) variant is often oligomeric. Thus, high protein concentrations could force even non-phosphorylated QseF to oligomerize (a well-known property of response regulators), which may impair its interaction with QseE and thereby also with RapZ.

As we are currently unable to provide an unambiguous explanation for the reduced response of RapZ to higher concentrations of the QseE:QseF mixture, we refrain from adding this novel experiment to the revised manuscript. To account for the referee's criticism, we included a sentence in the discussion stating that a simultaneous interaction of RapZ with QseE and QseF remains to be clarified (p. 15). Accordingly, we also slightly modified the model presented in Fig 7, avoiding to depict RapZ being simultaneously bound to QseE and QseF.

Referee #3:

In this study, Khan et al. set out to determine how cells are measuring intracellular flucosamine-6-phosphate (GlcN6P) to control the levels of the GlmY sRNA. The authors report that the RapZ RNA-binding protein itself binds GlcN6P and that the binding of GlcN6P controls the ability of RapZ to bind and activate the activity of the QseE-QseF two component system. Activated QseF response regulator activates *glmY* transcription. This forms a nice auto regulatory loop in which more GlmY titrates RapZ away from the GlmZ small RNA, leading to more GlmZ and then more GlmS protein and finally more GlcN6P, which in turn binds RapZ to prevent it from activating QseE-QseF as well as from binding GlmY.

The study includes a number of different approaches to test the proposed model, and the manuscript is carefully prepared.

I only have one major comment:

1. Since the concept that the RNA-binding protein RapZ also binds the GlcN6P small molecule as well as both the QseE sensor kinase and QseF response regulator is quite novel, I would love to see a little more analysis (mutational or crosslinking) to delineate the regions of the RapZ C-terminal domain that contact all of these different molecules.

We agree that further insight into the interaction surfaces used by RapZ would be of interest. We have already shown that the RapZ-CTD is sufficient for binding GlmY (Gonzalez et al., 2017) as well as GlcN6P (Figs 1C and 6D; this work). However, this does not apply to interaction with QseE/QseF as assumed by the reviewer. When tested by the bacterial two-hybrid assay, the separated C- and N-terminal domains of RapZ are not capable of efficiently binding QseF or QseE on their own (cf. graph A below; please note that corresponding RapZ BACTH constructs were previously used to demonstrate self-interaction of these domains confirming their functionality (Gonzalez et al., 2017)). Albeit the RapZ-CTD retains some interaction potential, this residual binding is not sufficient to activate *glmY* expression as shown by a complementation assay (cf. graph B below). Moreover, introduction of an Asp182Ala substitution into RapZ abrogating self-interaction of the C-terminal domain (as demonstrated previously in Gonzalez et al., 2017) also abolishes interaction with QseE as well as QseF (cf. columns 4 and 8 in graph A below). We conclude from these results (I) that both NTD and CTD of RapZ contribute to binding and activation of QseE/QseF and (II) that proper oligomerization of RapZ is required for interaction with QseE/QseF. These observations make it difficult to delineate an interaction surface by mutational analysis as at least two different regions in RapZ appear to be involved. Moreover, it will be difficult to discriminate between exchanges that specifically abrogate interaction and those that impair multimerization of RapZ, impeding a potential dissection of the interaction surface through a mutational screen. We added these novel data to the manuscript (Fig EV1) and describe and discuss them on p. 8 and p. 15, respectively.

Cross-linking cannot record interaction with the metabolite and will reveal interacting regions in proteins only roughly as it is restricted to a few functional groups. Analysis of the presumptive GlcN6P binding pocket also requires more time as corresponding mutants have to be thoroughly checked for various properties to prevent premature conclusions. We have previously characterized interaction surfaces of other protein complexes, which required long-term work and resulted in publications on their own (cf. Mörk-Mörkenstein, *Mol Microbiol.*, 2017, 106(1):54-73; Dickmanns et al., *J Biol Chem.*, 2018, 293(16):5781-5792). Based on this experience, we believe that delineating the interaction surfaces in RapZ is a follow-up question that must be carefully addressed in an own project in the future.

More minor comments:

2. Figure 6E: This EMSA image is not ideal.

We replaced the previous experiment with a better result.

3. Since the regulatory loop and the details of some of the experiments are challenging to keep track of, the writing needs to be as clear as possible. Two sentences that were confusing to me: -Page 5, line 5 from bottom: "Importantly, Nva-FMDP had no role for GImY* in the ΔrapZ mutant."

We changed the sentence accordingly: "Importantly, Nva-FMDP did not trigger accumulation of GImY* in the ΔrapZ mutant".

-Page 11, line 1 from top: "To survey these results,..."

Sentence changed to: "To verify these results..."

-Page 14, line 5 from top: "In this study, we identify the RBP RapZ..." (this sentence seems incomplete).

Sentence changed to: "In this study, we identify the RBP RapZ being at the heart of..."

4. Suggestions:

-Page 10, line 7 from bottom: Suggest "is capable of limiting its own".

O.K.

-Figure 1C: check heading of bottom panel.

Corrected.

-Figures 3, 4 and 6: Suggest adding lane numbers to figures, since the authors refer to specific lanes in the legend.

We added lane numbers as requested.

2nd Editorial Decision

20th Jan 2020

Thank you for implementing the final revisions in your manuscript. I apologise for the delay in communicating the decision due to the post-holiday backlog. To my assessment the minor issues indicated by the reviewers have now been addressed and there now remain only a few editorial issues that have to be finalised before I can extend formal acceptance of the manuscript.

2nd Revision - authors' response

21st Jan 2020

The authors performed the requested editorial changes.

3rd Editorial Decision

24th Jan 2020

Editor accepted the manuscript.

Corresponding Author Name: Boris Görke

Journal Submitted to: EMBO J.

Manuscript Number: EMBOJ-2019-103848